# Gut microbiota of an Amazonian fish in a heterogeneous riverscape: integrating genotype, environment, and parasitic infections

Nicolas Leroux,[1,2] Francois-Etienne Sylvain,[1,2] Aleicia Holland,[3] Adalberto Luis Val,[4] Nicolas Derome[1,2]

**ABSTRACT**  A number of key factors can structure the gut microbiota of fish such as environment, diet, health state, and genotype. *Mesonauta festivus*, an Amazonian cichlid, is a relevant model organism to study the relative contribution of these factors on the community structure of fish gut microbiota. *M. festivus* has well-studied genetic populations and thrives in rivers with drastically divergent physicochemical characteristics. Here, we collected 167 fish from 12 study sites and used 16S and 18S rRNA metabarcoding approaches to characterize the gut microbiome structure of *M. festivus*. These data sets were analyzed in light of the host fish genotypes (genotyping-by-sequencing) and an extensive characterization of environmental physico-chemical parameters. We explored the relative contribution of environmental dissimilarity, the presence of parasitic taxa, and phylogenetic relatedness on structuring the gut microbiota. We documented occurrences of *Nyctotherus* sp. infecting a fish and linked its presence to a dysbiosis of the host gut microbiota. Moreover, we detected the presence of helminths which had a minor impact on the gut microbiota of their host. In addition, our results support a higher impact of the phylogenetic relatedness between fish rather than environmental similarity between sites of study on structuring the gut microbiota for this Amazonian cichlid. Our study in a heterogeneous riverscape integrates a wide range of factors known to structure fish gut microbiomes. It significantly improves understanding of the complex relationship between fish, their parasites, their microbiota, and the environment.

**IMPORTANCE**  The gut microbiota is known to play important roles in its host immunity, metabolism, and comportment. Its taxonomic composition is modulated by a complex interplay of factors that are hard to study simultaneously in natural systems. *Mesonauta festivus*, an Amazonian cichlid, is an interesting model to simultaneously study the influence of multiple variables on the gut microbiota. In this study, we explored the relative contribution of the environmental conditions, the presence of parasitic infections, and the genotype of the host on structuring the gut microbiota of *M. festivus* in Amazonia. Our results highlighted infections by a parasitic ciliate that caused a disruption of the gut microbiota and by parasitic worms that had a low impact on the microbiota. Finally, our results support a higher impact of the genotype than the environment on structuring the microbiota for this fish. These findings significantly improve understanding of the complex relationship among fish, their parasites, their microbiota, and the environment.

**KEYWORDS**  microbial ecology, host-parasite relationship, gut microbiota, fish pathogens, environmental microbiology, Amazonia, metabarcoding, population genetics

Address correspondence to Nicolas Leroux, nicolas.leroux.1@ulaval.ca.

The authors declare no conflict of interest.

See the funding table on p. 20.

Fish gut microbiota is now acknowledged to play important roles in their host's metabolism, immunity, and even behavior (1–4). The taxonomic composition and functional repertoire of the gut microbiota are determined by a wide range of factors. From these, four usually stand out as the most important: environmental conditions, host diet, genotype, and physiological condition (i.e., developmental stage and state of health) (2, 5–7). The contribution of these factors on the structure of the gut microbiota has been addressed previously using *in vivo* experiments on the zebrafish model, *Danio rerio* (8–10). However, modeling *in situ* host-microbiota interaction with *in vivo* experimental data is far from being predictive since *in situ* studies involve a much more complex interplay of multiple factors. The complexity of field studies makes it extremely important for *in situ* studies to have both elaborate sampling designs and extensive knowledge of the species to allow generation of meaningful results.

The environment is one of the main factors shaping the gut microbiota of fish (11). Fish are in permanent contact with the surrounding water's microbial communities, which can act as the primary source of fish microbial symbionts (12, 13). By consequence, changes in water physicochemical parameters or other environmental alterations (e.g., xenobiotics) affect the taxonomic structure of the gut microbiota of fish (14–17). The Amazon Basin is characterized by heterogeneous aquatic ecosystems resulting from two main water types with very contrasting chemistries (18), fish communities (19) and water microbial communities (20). White water systems are characterized by a cloudy appearance caused by a larger amount of suspended solids, a lower amount of dissolved organic carbon (DOC), a circumneutral pH, and a higher quantity of ions than black water, which is rich in DOC, acidic, and ion poor. For instance, conductivity at 25°C for black water is close to 8 µS/cm versus 70 µS/cm for white water. In addition, the pH of black water systems is usually lower than 5, compared to 7 for white water (21–24). The bacterioplankton communities from these different water types also significantly differ and are correlated with differences in environmental parameters (25). Since the surrounding water may be a source of recruitment of environmental microbial strains to the fish gut microbiota (13, 15), we posed the hypothesis that certain taxa of *Mesonauta festivus*' gut microbiome would be related to the water type (black or white) and potentially recruited by horizontal transfer. If environment plays a major structuring role on microbial communities, the microbiota of a given fish species living in both water types should harbor discriminant taxonomic features specific to each environment.

Fish microbiota can also be associated with the physiological condition of their host. The gut microbiota is an important structuring component of the innate immunity of fish (26) and is in direct contact with parasites colonizing the gut. Moreover, several fish pathologies have been associated to gut microbiota dysbiosis (i.e., imbalance) (27–29). For instance, Llewellyn et al. (30) documented a disruption of the skin microbiome of the Atlantic salmon following infection with the parasitic copepod *Lepeophtheirus salmonis*, favoring prevalence of opportunistic bacterial strains at the expense of putative commensals (30). Disruption of microbiome following parasitic infection was also confirmed in a zebrafish model. More recently, both synergistic and antagonistic interactions between ciliates and the microbiome have been described (31, 32). Therefore, the microbiota of wild fish infected by certain types of parasites could also present signs of dysbiosis involving parasite-specific taxonomic features. Given that parasitism of fish exerts tremendous negative effects on the economy and health of Amazonian riverine communities (33), many studies explored the complex dynamic of parasitic infections in Amazonia. To date, all of them have focused on visual identification, rather than using molecular approaches such as metabarcoding, thus limiting both the exhaustivity and accuracy of parasite detection and identification (34–37). A more extensive characterization of the parasitic diversity present in each Amazonian water type and its interactions with the gut microbiota of wild fish is, therefore, needed. The use of next-generation sequencing will provide improved data and may lead to a more sophisticated statistical framework for further research on this matter.

Host phylogenetic relatedness is another key determinant of the gut microbiota taxonomic structure (38–40). Indeed, close relationships between fingerlings and parents in species that exert extensive parental care (e.g., Cichlidae and Gastrerosteidae) should favor vertical transfers, ultimately leading to akin microbiotas in phylogenetically related fish (12). In the same way, similar genetic backgrounds could favor the recruitment of similar microbiomes in closely related fish. For instance, Smith et al. (41) observed a higher gut microbiota dissimilarity between stickleback populations that are more genetically divergent. According to this, if host phylogenetic relatedness is an important determinant of the gut microbiota taxonomic structure, fish from the same genetic population should harbor similar gut microbiomes when considering their water type of origin and the presence of parasitic infections.

However, differentiating the effects of the phylogenetic relatedness and the environment is a complex challenge in natural systems, as high environmental dissimilarity between sites sometimes indirectly leads to a stronger genetic differentiation (42). Considering the high environmental heterogeneity of the Amazonian watershed, deciphering genetic and environmental effect requires the comparison of multiple connected black and white water sites involving different genetic populations. In any case, the interplay between genetic and environmental effect on gut microbiota taxonomic structure ranges between the following extreme scenarios: strong environmental effects relative to weak genetic effects and vice versa. In the first case, fish should display similar microbial community shifts at similar environmental shifts (e.g., independent black water sites versus their respective connected white water sites). On the opposite, in the presence of a weak environmental effect and a strong genetic effect, fish from a same genetic population, but inhabiting contrasting environments, are expected to converge in terms of microbiota composition.

This study aimed to understand how the gut microbiome of *M. festivus* is structured by their hosts' phylogenetic relatedness, parasitic communities, and the environment. Here, we combined 16S and 18S rRNA metabarcoding approaches to characterize the gut bacterial and eukaryotic communities from four genetic populations of *M. festivus* distributed across a wide range of rivers (black and white) within the Brazilian Amazon basin. We described (i) the host-parasite prevalence rate in black and white water environments, (ii) the possible roles that environment and phylogenetic relatedness play on structuring the gut microbiota, and (iii) the effect of gut dwelling taxa presence in samples on the gut microbiome. *M. festivus*, a detritivore cichlid ubiquitous to the Amazon basin, was chosen as the model organism since it has a high environmental tolerance to environmental variations such as drastically contrasting abiotic conditions between black and white waters (43). In addition, parasitic infections by nematodes have previously been reported in wild specimens of *M. festivus* (44), and the species has important parental care investment, staying with fingerlings post hatching (43). Finally, in terms of genetic diversity, the four populations of *M. festivus* recorded in Amazonia are not structured according to the water physicochemical characteristics (45), therefore, making this species model suitable to disentangle environmental versus genetic effect on microbiota composition. Taken together, *M. festivus* is a relevant model to measure the relative contribution of environment, genotype, and parasitic infection in shaping the taxonomic structure of fish microbiota in a natural system.

## MATERIALS AND METHODS

### Experimental design

We sampled 167 *M. festivus* at 12 sites distributed across three major Amazonian rivers (i.e., Rio Branco, Rio Negro, and Rio Solimoēs) (permit numbers 2018021-1 and 29837-18) (Fig. 1). These sites comprised ecosystems with drastically divergent physicochemical parameters: five black water sites (BAR, NEG, CEM, ANA, and TEF) and seven white water sites (PIR, SOL, MAN, JAR, JAC, CAT, and BRA) (Table 1). Field trips were conducted from September to December 2018–2019 during the dry season. We sampled every

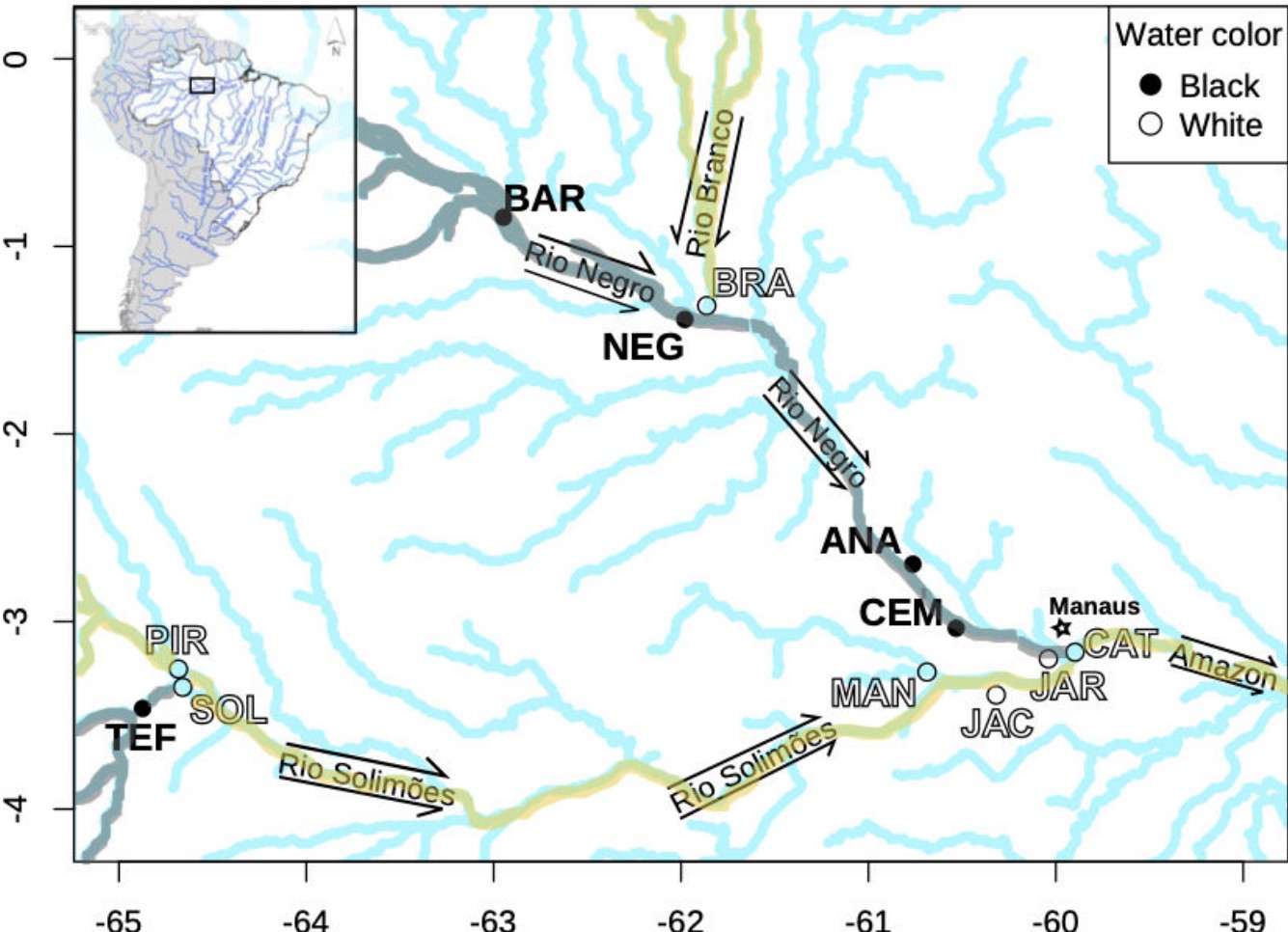

**FIG 1** Location of sampling sites (*n* = 12) in the Amazon basin. The three major tributaries of the Amazon are labeled, and the water flow directions are identified using arrows. Water is always flowing toward the East, in the direction of the Amazon. Sites have been identified using their three-letter acronyms, and point colors are consistent with the described water type at a given site. Main rivers have been colored according to their water type [modified from reference (49)].

fish during the same season as there are parasites which prevalence is known to be more variable between seasons in Amazonia (46, 47). We estimated the watercourse distance separating each sampling site using Google Earth pro. A multiparameter YSI Professional + Series meter (YSI Inc/Xylem Inc., USA) was used to characterize water physicochemical properties (conductivity and pH) at each site. Two liters of water were also sampled at each site 30 cm below the surface, the depth where *M. festivus* were fished (43), to characterize other water parameters in the laboratory. At the laboratory, DOC was quantified and characterized, and the concentrations of nutrients ($NO_2^-$, $NO_3^-$, and silicates), free ions (Ca, Na, Cl, Mg, and K), and 12 metals (Al, V, Cr, Mn, Fe, Co, Ni, Cu, Zn, As, Cd, and Pb) were determined according to the method detailed in Sylvain et al. (25). Fish specimens were collected at each sampling site using a combination of small seine net fishing and line fishing (fish collection permit number 29837-18). Shortly after collection, fish were euthanized using a classic MS-222 protocol to dissect a fin clip, a gill, gut tissues, and stomach content. All tissues were kept individually in Nucleic Acid Preservation (NAP) buffer to preserve DNA integrity (48). Samples were frozen at −20°C right after the dissection and until DNA extraction.

**TABLE 1**  Water type at each site, genetic population of origin (genotype), DOC concentration, ionic composition (Na, Mg, K, Ca, and Cl), nutrient concentration (silicate), productivity (chlorophyll a: Chl a), physiochemical characteristics (conductivity: Cond and pH), and global positioning system (GPS) coordinates (GPS S and GPS W) measured at each sampling site ($n = 12$)[a]

| Site | ID | n | Color | Genotype | DOC | Na | Mg | K | Ca | Cl | Silicate | Chl a | Cond. | pH | GPS S | GPS W |
|---|---|---|---|---|---|---|---|---|---|---|---|---|---|---|---|---|
| Barcelos | BAR | 12 | Black | A | 10.93 | 0.46 | 0.12 | 0.42 | 0.04 | 0.11 | 64 | 0.35 | 13.1 | 3.71 | 0°50'50.8 | 62°57'40.3 |
| Rio Negro | NEG | 10 | Black | A | 11.67 | 0.25 | 0.09 | 0.33 | 0.49 | 1.16 | 92 | 0.73 | 10.6 | 4.16 | 1°23'29.8 | 61°59'35.3 |
| Rio Branco | BRA | 18 | White | B | 6.04 | 1.15 | 0.43 | 0.7 | 0.93 | 1.1 | 180 | 6.21 | 22 | 6.25 | 1°19'05.7 | 61°52'34.7 |
| Anavilhanas | ANA | 13 | Black | B | 11.38 | 1.8 | 0.26 | 0.65 | 0.08 | 0.32 | 73 | 0.05 | 13.2 | 4.24 | 2°41'46.1 | 60°46'33.3 |
| Lago do cemitério | CEM | 8 | Black | B | 9.76 | 0.23 | 0.05 | 0.14 | 0.37 | 0.64 | 77 | 1.35 | 7.2 | 3.83 | 3°02'16.6 | 60°32'42.7 |
| Janauari | JAR | 16 | White | C | 7.13 | 1.99 | 0.2 | 0.79 | 0.06 | 1.47 | 98 | 4.62 | 22.4 | 4.38 | 3°12'03.4 | 60°03'10.1 |
| Catalão | CAT | 17 | White | C | 9.05 | 4.56 | 3.76 | 1.71 | 0.83 | 1.75 | 242 | 7.14 | 174.8 | 5.7 | 3°09'56.4 | 59°54'38.4 |
| Lago Janauacá | JAC | 17 | White | C | 5.73 | 3.32 | 1 | 1.07 | 0.44 | 2.17 | 157 | 1.35 | 88 | 6.75 | 3°23'37.5 | 60°19'52.6 |
| Manacapuru | MAN | 12 | White | C | 7.97 | 4.91 | 0.14 | 1.45 | 0.05 | 1.43 | 126 | 2.78 | 24.3 | 5.31 | 3°16'16.9 | 60°42'03.2 |
| Téfé-Solimões | SOL | 9 | White | D | 5.73 | 1.95 | 0.21 | 0.28 | 1.11 | 1.29 | 327 | 4.41 | 19.7 | 6.05 | 3°21'07.4 | 64°40'21.4 |
| Lago Tefé | TEF | 17 | Black | D | 7.13 | 0.87 | 0.19 | 0.56 | 0.82 | 0.53 | 218 | 1.82 | 10.6 | 4.98 | 3°27'55.2 | 64°53'13.2 |
| Lago Pirates | PIR | 18 | White | C | 6.47 | 5.35 | 1.76 | 1.28 | 1.17 | 3.26 | 222 | 9.05 | 127.6 | 7.15 | 3°15'19.2 | 64°41'44.3 |

[a]Values are the average of three readings. Units are in mg/L for DOC and ions, µmol/L for silicate, µg/L for productivity, and µS/cm for conductivity. $n$ = number of fish sampled at the given site.

## Next-generation sequencing library preparation

A total of 167 midgut samples were collected and processed. DNA was extracted from 20 mg of a midgut segment for each fish. The middle section of the intestine was specifically selected, and samples were cut into small pieces using sterile blades and digested with lysis buffer overnight. Although an additional step of mechanical disruption would have improved the detection of both microbial and parasitic taxonomic diversity, this potential bias should not affect comparisons between experimental groups. Genomic DNA was extracted using the QIAGEN DNeasy Blood and Tissues kit following the manufacturer's instructions. DNA quality and quantity were assessed with a NanoDrop 2000 spectrophotometer. For 18S rRNA metabarcoding, the V4-V5 regions of the 18S rRNA gene were amplified by polymerase chain reaction (PCR) using the amplification primers 616*F and 1132R, a combination of primers designed to optimize parasite species detection (50). Two elongation arrest blocking primers, 816F-C3 and 846R-C3, modified with a C3 spacer to specifically inhibit the amplification of host DNA were added to the reaction according to the protocol described in reference (51). The following program was used: (i) 30 s at 98℃, (ii) 10 s at 98℃, (iii) 30 s at 54℃, (iv) 20 s at 72℃, and (v) 2 min at 72℃, with 35 amplification cycles. For 16S rRNA metabarcoding, the V3-V4 regions of the 16S rRNA gene were amplified by PCR using the amplification primers 341F and 805R. The following program was used: (i) 30 s at 98℃, (ii) 10 s at 98℃, (iii) 30 s at 60℃, (iv) 20 s at 72℃, and (v) 2 min at 72℃, with 35 amplification cycles. After the first amplifying PCR, both the 16S and 18S rRNA next-generation sequencing libraries followed the same protocol. We ran the PCR product on a 2% electrophoresis gel to assess the efficacy of the PCR and purified the amplified DNA with AMPure beads (Beckman Coulter Genomics) to eliminate shorter sequences, primer dimers, proteins, and phenols. Post‐PCR-purified DNA concentration and quality were assessed on Nanodrop 2000. Indexing PCRs were completed with the following program: (i) 30 s at 98℃, (ii) 10 s at 98℃, (iii) 30 s at 60℃, (iv) 20 s at 72℃, and (v) 2 min at 72℃, with 15 amplification cycles. We ran the PCR product on a 2% electrophoresis gel to assess the efficacy of the PCR, purified the amplified DNA with AMPure beads (Beckman Coulter Genomics), and then verified the DNA concentration and quality on Nanodrop 2000. Both positive and negative controls were included with each PCR (one per PCR plate) for each step of the PCR amplifications ($n = 8$). None of the negative controls amplified during PCRs. Multiplex paired-end sequencing was performed using the MiSeq platform from Illumina (Illumina), by the Plateforme d'analyses génomiques at the Institut de Biologie Intégrative et des Systèmes (IBIS) at Laval University using

the MiSeq Reagent Kit v3. All proceedings followed the manufacturer's protocols. All negative controls were sequenced, but no complete sequence was retrieved, supporting the absence of contamination during PCR amplification.

## Processing of high-throughput sequencing data

The analysis of amplicon sequences was done at the IBIS at Laval University. After sequencing and removing primers with CutAdapt (52), 13,432,629 and 10,338,365 sequences were, respectively, obtained from 16S and 18S libraries sequencing, respectively, leading to a mean of 65,525 and 57,756 sequences per sample. The demultiplexed fastq sequence files were processed through DADA2 (53) using the function "ytfilterAndTrim" with the following parameters: two as the phred score threshold for total read removal, a maximum expected error of two for forward reads and four for reverse reads, a truncation length of 280 base pairs for forward reads and 200 base pairs for reverse reads for 16S rRNA library, and a length of 275 base pairs for forward reads and 250 base pairs for reverse reads for 18S rRNA library. The filtered reads were then fed to the error rate learning, dereplication, and amplicon sequence variant (ASV) inference steps using the functions "learnErrors", "derepFastq", "*mergePairs,*" and "dada." Chimeric sequences were removed using the "removeBimeraDenovo" function with the "pseudo" method parameter. For 16S rRNA, taxonomic annotation was done using the function "assignTaxonomy" on the Silva rRNA SSU Ref NR 99 Release 138.1. For 18S rRNA, taxonomic annotation was done using *blastn* matches against the NCBI refseq SSU 18S database, whereas all hits above 90% identity were considered to determine the taxonomy using a lowest common ancestor algorithm in a manner inspired by MEGAN (54). *Blastn* matches with 99% coverage and 99% identity were assigned the match's taxonomic identity. Since the number of 18S rRNA ASVs was low, we manually curated the 18S rRNA taxonomic table to reduce the chances of wrongfully classifying an ASV. We verified that the sequencing depth of samples was sufficient for both marker genes using rarefaction analyses. We calculated taxonomic trees based on the DNA sequences using the *R* package *ape* (55, 56), and a Phyloseq (57) object was produced for each library in *R* to do subsequent analyses. In the 18S rRNA Phyloseq object, we removed ASVs with fewer than two read counts or without a taxonomic assignation at the kingdom level. For the 16S rRNA Phyloseq object, we removed ASVs with fewer than 10 read counts or without a taxonomic assignation at the phylum level.

## Statistical analyses

### Estimating the parasitism rate in both water types

Using the 18S rRNA *Phyloseq* object previously produced, we removed every ASVs assigned to the Class *Actinopterygii* from the data set with the function "subset_taxa" from *Phyloseq* to remove host ASVs. We manually investigated the data set for the repeated presence of parasitic taxa and detected the presence of six taxonomic groups of parasites of global interest for our study [i.e., *Nyctotherus* sp., Nematoda, Platyhelminthes (Cestoda and Trematoda), Acanthocephala, and Microsporidia]. We did not quantify the abundance of parasitic DNA observed in a given sample as our method rather aim at detecting eukaryotic parasite taxa. Indeed, the amplification bias caused by the variable amount of host tissues and the amount of food bolus present in the gut would have biased the quantification. Instead, we used a binary (yes or no) statistic to describe the presence or absence of a given parasite in a gut sample. A fish with more than five sequences from an ASV assigned to a given parasitic taxa was considered as a fish with the parasite present in his gut. Likewise, the detection of a parasite in a gut sample does not mean the fish is infected, as DNA-based methodologies can detect both living and dead parasites originating from difference sources.

Only three types of parasites had a sufficiently high presence rate for us to produce statistical analyses (i.e., *Nyctotherus* sp., Nematoda, and Platyhelminthes). We used chi-square tests of independence with Yates' continuity correction, using the function

"chisq.test" to look for a dependency between the water type and the three types of parasite infection studied. We also estimated the mean prevalence of these taxa at each site and calculated a mean for each water type to produce a boxplot representing the mean prevalence of *Nyctotherus* sp., Nematoda, and Platyhelminthes in both water types.

### Describing gut microbiota in both water types

Using the 16S rRNA Phyloseq object previously produced, we estimated alpha diversity indexes, such as the observed number of ASVs (Chao1), and Shannon entropy (58) using estimate_richness function. Then, phylogenetic diversity (59) was computed with using estimate_pd(phylo) function. The tree was computed with the simple agglomerative (bottom-up) hierarchical clustering method - Unweighted Pair Group Method with Arithmetic Mean (UPGMA). We verified for differences between the mean alpha diversity indexes from fish gut samples in both water types using a two-sided, two-sample Wilcoxon test with the function "wilcox.test." We used a non-parametric statistical test as the number of fish at black ($n$ = 60) and white ($n$ = 107) water sites was not equal and diversity indexes were not normally distributed for all groups. Then, we computed a permutational multivariate analysis of variance (PERMANOVA) based on Bray-Curtis beta diversity distances, using the function "adonis2" from *Vegan* (60) in R with 200,000 permutations, considering the 16S rRNA ASV table as the response variable. The water types and genetic population of each sample were the explicative variables. There was no interaction between explicative variables. Leroux et al. (2022) already genotyped *M. festivus* samples at these 12 sites and found 4 distinct genetic populations, which we considered in our analysis. We then plotted the results of the PERMANOVA using a non-metric multidimensional scaling (NMDS) ordination approach based on both the Bray-Curtis and weighted UniFrac indexes. Since both distance indexes led to similar results, we only included the weighted Bray-Curtis index in our main results.

We used the function "envfit" from *Vegan*, with 1,000 permutations, considering five physicochemical parameters documented as major differentiating factors between black and white water environments as explicative variables (i.e., dissolved organic carbonate, silicate in suspension, productivity rates, conductivity, and pH) (23). We opted for an ordination-based method instead of a PERMANOVA, as it allows for the integration of environmental vectors and beta-diversity data within a graphical output, thereby highlighting the relative influence of each parameter on the gut microbiome structure. Although this analysis may be comparatively less powerful in detecting significant associations, it yields more interpretable and scientifically significant results. We fitted the five environmental predictors onto the ordination analysis as vectors to show how these five environmental variables correlate with the Bray-Curtis dissimilarity index of the midgut microbiota of *M. festivus*.

We conducted a linear discriminant analysis effect size (LEfSe) (61) to identify taxa mainly characterizing the differences between the midgut microbiota of *M. festivus* in black and white water environments. We used a 16S rRNA Phyloseq object agglomerated by genus, using the function "tax_glom" from Phyloseq, as the response variable and the water type where the fish of each midgut sample was collected as the explicative variable. For the parameters, we used a value of $1 \times 10^{-3}$ for the alpha value of the factorial Kruskal-Wallis test among classes and a threshold of 3 on the logarithmic Linear Discriminant Analysis (LDA) score for discriminative features. These parameters are very stringent and aim at only discovering highly discriminative features, lowering the potential for false positives. We used the "Plot LEfSe Results" and "Plot Cladogram" features of the Galaxy tool to plot the results of the analysis. To compare with the LEfSe analysis, we filtered the 25 most abundant genera from the 16S rRNA Phyloseq object using the function "prune_taxa" from the package Phyloseq. Then, we merged samples according to their water type and produced relative abundance bar plots at the class taxonomic resolution to look for differences in abundance between the midgut microbiota of *M. festivus* in both water types.

## Describing interactions between parasites and the gut microbiota

Since we did not detect a significant or important effect of the detection of any kind of helminth [i.e., Nematoda, Platyhelminthes (Cestoda and Trematoda) and Acanthocephala] on the structure of the midgut microbiota of *M. festivus*, we pooled all the presence of these parasites into the "fish with helminth" category. We previously analyzed independently the impact of each type of helminth detected, and their pooling did not have any impact on the general significance of the results. The other taxon of interest is the ciliate genus *Nyctotherus* sp., for which we found a high infection rate. We conducted separate analyses for both types of parasite (Helminths and *Nyctotherus* sp.) while considering cases of multiple infection by both types of parasite in the same sample. We did not detect coinfection pattern between the infection rates of any parasite detected.

Using the 16S rRNA Phyloseq object, we estimated the total number of ASVs (observed diversity), the Faith's phylogenetic diversity, and the Shannon alpha diversity indexes for each fish midgut sample. We also estimated the total number of ASVs from the phylum Proteobacteria in samples. Then, we grouped samples according to the detection or not of ciliates of the genus *Nyctotherus* sp. and helminths in the 18S rRNA data set. Since the design of the experiment was not balanced, both for the helminths and *Nyctotherus* sp., and some diversity indexes were not normally distributed for all groups, we used two-sided, two-sample Wilcoxon tests, using the function "wilcox.test," to look for differences between the mean alpha diversity index of each group. We also computed multiple-factor PERMANOVA based on Bray-Curtis beta diversity distances, using the function "adonis2" from *Vegan* in R with 50,000 permutations, considering the genera of the 16S rRNA ASV table as the response variable. We considered the presence of helminths, the presence of *Nyctotherus* sp., the water type of provenance, and the genetic population for each sample as the four explicative variables. Considering that the Adonis function is often influenced by the order of regression covariates, we conducted multiple runs of the Adonis function, employing various covariate orders. The outcome of the statistical test remained unaffected regardless of the order of covariates. There was no significant interaction between explicative variables, except between the genotype and the presence of *Nyctotherus* sp. ($R^2 = 0.03$, *P*-value = 0.003). However, considering the genetic population and the presence of *Nyctotherus* sp. independently leads to the same results. We plotted the results of the PERMANOVA using an NMDS based on both the Bray-Curtis and weighted UniFrac indexes for both types of infection. The stress values were 0.24 and 0.18 for Bray-Curtis and weighted Unifrac indexes, respectively. Since both distance indexes led to similar results, we only included the Bray-Curtis index in our main results. Then, we filtered the 50 genera with the highest relative abundance from the 16S rRNA Phyloseq object using the function "prune_taxa" from the package Phyloseq. We merged samples according to the presence or absence of *Nyctotherus* sp. and helminths and produced relative abundance bar plots at the class taxonomic resolution to look for differences in relative abundance between the midgut microbiota of *M. festivus* in each group for each parasite type.

We produced a microbiota co-abundance network for samples in which we detected infection by *Nyctotherus* sp. and a co-abundance network for fish uninfected by *Nyctotherus* sp. We used the CoNet application (62) in Cytoscape (63) using the number of sequences (abundance) of the 100 most abundant genera present in the 16S rRNA Phyloseq object agglomerated at the genus level as the data matrix. The following parameters were selected according to the CoNet documentation on 16S V35 phylotypes data subset (62): for the randomization and the bootstrap, we activated parent-child exclusion, filtered the data matrix with a minimum occurrence of 10 in rows, standardized the data using a column normalization, activated the Pearson and Spearman correlation indexes, activated the mutual information parameter, activated the Bray-Curtis and Kullback-Leibler dissimilarity distances, chose an automatic threshold setting edge selection parameter of 100, chose a randomization routine of 100 iterations based on the edgeScores, and selected the brown *P*-value merging. We formatted both

## RESULTS

### *M. festivus* gut microbiota in both water types

#### Eukaryotic communities

The phyla Ciliophora, Nematoda, Streptophyta, and Platyhelminthes were the most abundant out of the 135 ASVs detected from the 18S rRNA data set. Within this, six classes of gut dwelling taxa were detected: Chromadorea, Cestoda, Trematoda, Neoechinorhynchida, Amorphorea, and Microsporidia. Only gut dwelling Nematodes, Platyhelminthes, and the ciliate *Nyctotherus* sp. were detected in sufficiently high number of fish for us to complete meaningful statistical analyses. We detected *Nyctotherus* sp. in 53 samples, compared to 21 and 26 for nematodes and platyhelminths, respectively (Fig. 2). We detected Platyhelminthes from the classes Cestoda ($n = 24$) and Trematoda ($n = 4$). In two occurrences, we detected both a cestode and a trematode in the same fish. Likewise, 87 samples were free from these taxa. According to the Pearson chi-square test comparing the prevalence of these three gut dwelling taxa in both water types, the prevalence of Platyhelminthes was significantly associated to the water type ($X$-squared = 12.167, df = 1, and $P$-value = $4.8 \times 10^{-4}$) (Fig. 3). Conversely, *Nyctotherus* sp. ($X$-squared = 0.25527, df = 1, and $P$-value = 0.6134) and nematodes ($X$-squared = 0.21584, df = 1, and $P$-value = 0.6422) were independent to water type. We only detected one infection by a platyhelminth in samples from black water ($n = 60$), while there were 25 occurrences in samples from white water ($n = 107$).

#### Bacterial communities

Alpha diversity metrics (i.e., observed, Faith's phylogenetic, and Shannon) of *M. festivus* midgut microbial communities were not significantly different in gut samples collected from different water types according to the two-sided Wilcoxon tests ($P$-values >0.6). In the "Envfit" analysis, there was a significant correlation between the Bray-Curtis dissimilarity indexes and the chlorophyll concentration in water ($R^2 = 0.084$, $P$-value = 0.001), while the four other physicochemical parameters tested were not significantly correlated with the beta diversity (DOC: $R^2 = 0.003$, $P$-value = 0.76; silicate: $R^2 = 0.027$, $P$-value = 0.11; conductivity: $R^2 = 0.027$, $P$-value = 0.11; pH: $R^2 = 0.024$, $P$-value = 0.14). According to the NMDS based on Bray-Curtis dissimilarity index fitted with environmental vectors, microbiome from fish sampled at white water sites is structuring in the same axis as the productivity (Fig. 4). In the PERMANOVA based on Bray-Curtis dissimilarity index, the population genotype had a more important effect than water type on fish gut microbiome (genetic: $R^2 = 0.04$, $P$-value = $1.5 \times 10^{-5}$; water type: $R^2 = 0.02$, $P$-value = $5.0 \times 10^{-6}$).

The LEfSe analysis identified discriminative features from seven different taxonomic classes present in gut samples from black and white water sites (Fig. 5A). The classes Gammaproteobacteria, Brevinematia, Cyanobacteria, and two families from Acidimicrobiia and Deinococci were associated with guts from white water sites. Similarly, Acidobacteriae and family of Clostridia were associated with guts from the black water environment. In the barplots showing the relative abundance of the 25 most abundant genera (Fig. 5B), we observed a higher relative abundance of genera within the classes Alphaproteobacteria, Brevinematia, Bacteroidia, and Gammaproteobacteria in samples from white water, while we observed a higher relative abundance of the classes Clostridia and Desulfovibrionia in samples from black water. From these results, we detected a strong concordance between the LEfSe analysis results and the bar plots relative abundances for three taxonomic classes (i.e., Gammaproteobacteria, Brevinematia, and Clostridia) (Fig. 5). However, as these taxa were not detected in the water microbiome sampled in each water type, it was not possible to link these discriminant features to a putative horizontal transfer from the water microbiome.

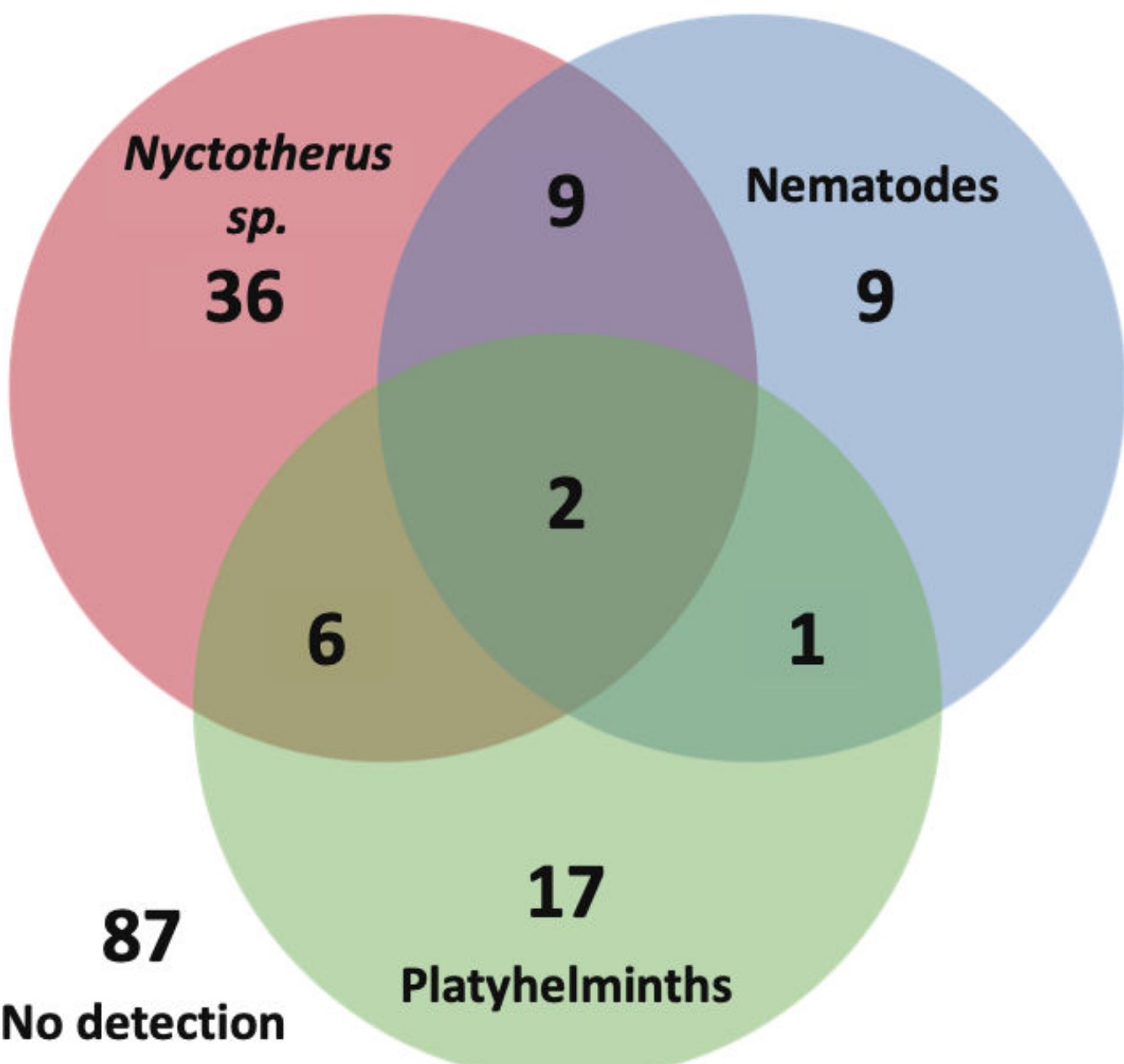

**FIG 2** Venn diagram representing the number of *M. festivus* midgut samples (*n* = 167) where we detected the presence of the ciliate *Nyctotherus* sp., nematodes, or platyhelminths using a 18S rRNA gene metabarcoding approach. In total, we detected 53 *Nyctotherus* sp. infections, compared to 21 and 26 for nematodes and platyhelminths, respectively. This Venn diagram was produced using the library VennDiagram in R.

### Influence of parasites on the gut microbiota

There were significant differences between the alpha diversity of the midgut microbiota of fish infected by *Nyctotherus* sp. in the two-sided Wilcoxon rank sum test. For the three diversity metrics considered, the alpha diversity was significantly higher in samples where *Nyctotherus* sp. was detected (observed diversity: $W$ = 3715.5, *P*-value = 0.01702; Faith's phylogenetic: $W$ = 4,078, *P*-value = 0.0003; Shannon diversity: $W$ = 3,910, *P*-value = 0.002251) (Fig. 6). In contrast, we did not detect any significant differences between mean alpha diversity indexes in fish where helminths were detected (observed diversity: $W$ = 2,421, *P*-value = 0.1953; Faith's phylogenetic: $W$ = 2,598, *P*-value = 0.5087; Shannon diversity: $W$ = 2,625, *P*-value = 0.5726). Furthermore, samples in which *Nyctotherus* sp.

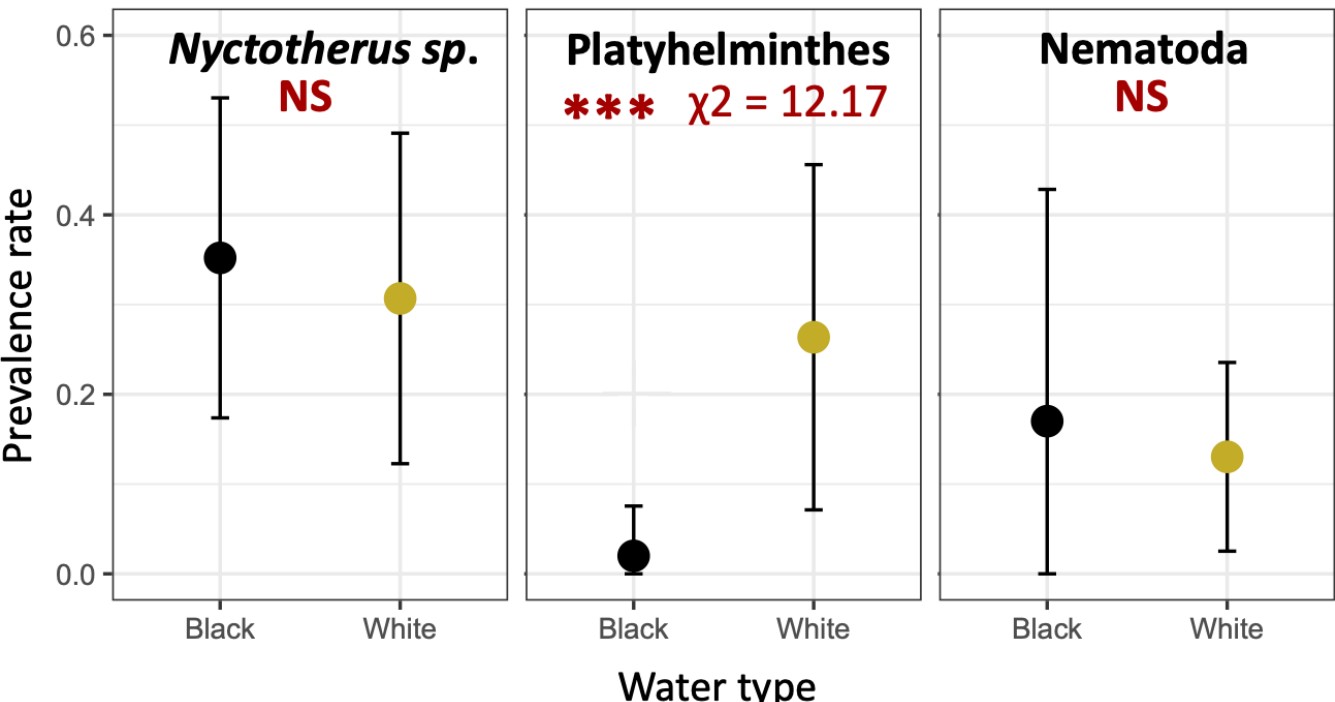

**FIG 3** Mean prevalence rate of *Nyctotherus* sp., Platyhelminthes, and Nematoda estimated at each site (*n* = 12) and clustered by water type for *M. festivus* midgut samples (*n* = 167). We included the results of the three chi-square tests of independence in red at the top of each plot. NS stands for not significant.

or helminths were detected had a significantly higher number of ASVs from the phylum Proteobacteria in their microbiome (*Nyctotherus* sp.: $W$ = 1748.5, $P$-value = $1.222 \times 10^{-05}$; Helminths: $W$ = 2,191, $P$-value = 0.03409) (Fig. 6).

A moderate influence of *Nyctotherus* sp. (PERMANOVA: $R^2$ = 0.05, $P$-value $<2 \times 10^{-05}$) and a low influence of the presence of helminths (PERMANOVA: $R^2$ = 0.01, $P$-value = 0.045) were detected on the taxonomic composition of *M. festivus* gut microbiome. In addition, the water type from which samples were collected had a low influence (PERMANOVA: $R^2$ = 0.02, $P$-value = $1.5 \times 10^{-3}$) on the Bray-Curtis dissimilarity index, while the genetic population of samples had a moderate influence (PERMANOVA: $R^2$ = 0.04, $P$-value $<1.2 \times 10^{-4}$) across the 12 sites. In the Principal Coordinate Analyses (PCoAs) based on Bray-Curtis dissimilarity index, we observed a structuration of samples infected by *Nyctotherus* sp., while there was no apparent difference detected in the presence of helminths (Fig. 7B). Furthermore, samples infected by *Nyctotherus* sp. had a lower relative abundance of Proteobacteria and hosted Halobacterota, which was absent from the samples without *Nyctotherus* sp. (Fig. 7A). On the contrary, the detection of helminths in samples was not associated to differences in the 50 most abundant genera.

The CoNet co-abundance networks show a restructuration of the midgut microbiota taxonomic structure of *M. festivus* associated with the presence of *Nyctotherus* sp. (Fig. 8). The network based on samples with *Nyctotherus* sp. included 23 genera of Proteobacteria, while the network based on other samples included only 16. In addition, Proteobacteria are responsible for 33% of edges (links) in the network based on samples infected by *Nyctotherus* sp. (47 out of 141 edges) in comparison to 14% for the network based on fish without the parasite (13 out of 92 edges). The two co-abundance networks have a similar total number of nodes, 69 and 64, respectively, but the network based on fish infected by *Nyctotherus* sp. had a higher number of edges (141 vs 92). The higher number of edges mainly results from the higher diversity and connectivity of Proteobacteria in the network. Interestingly, the network based on fish infected by

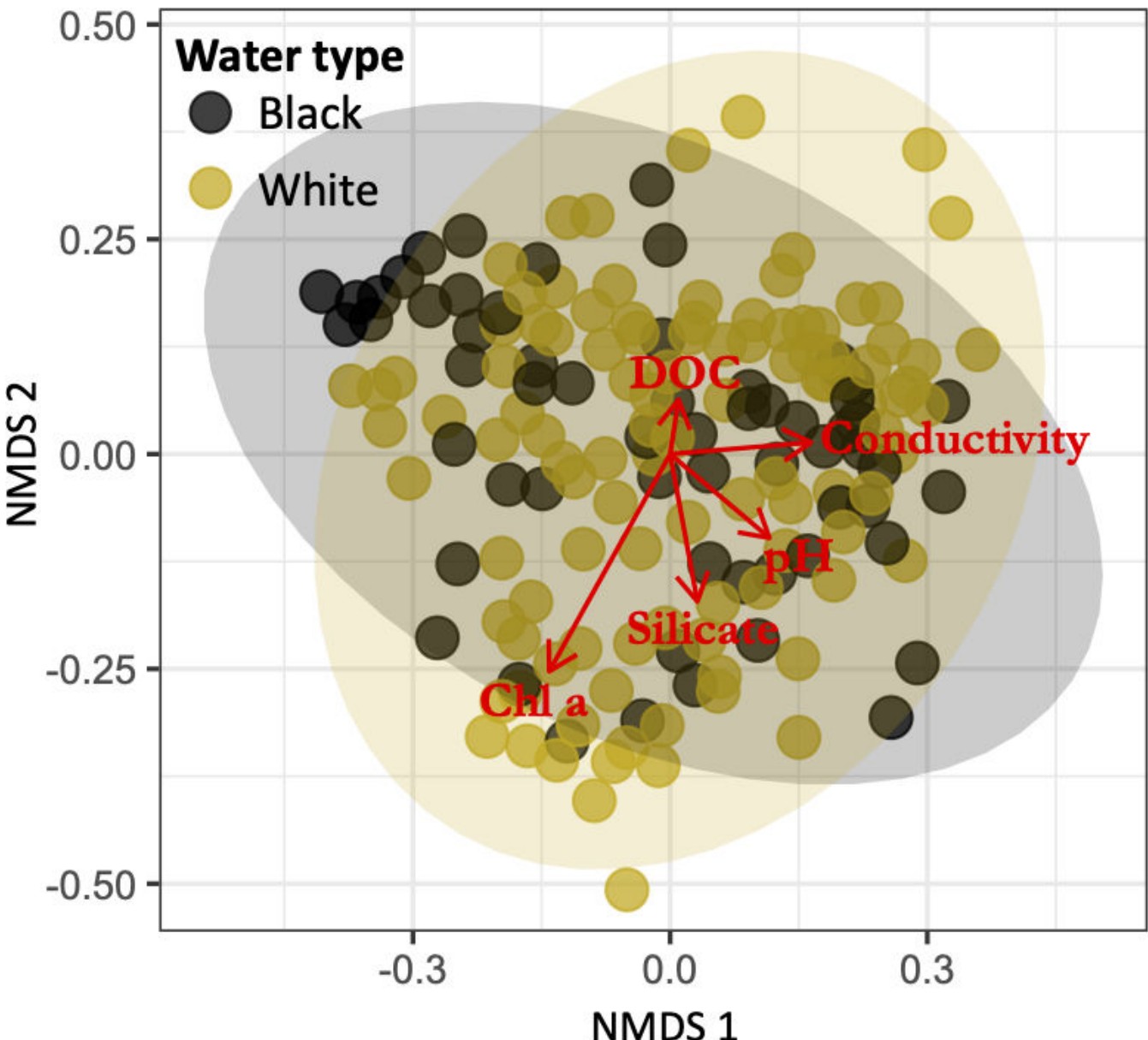

**FIG 4** Non-metric multidimensional scaling based on the Bray-Curtis dissimilarity index between the 16S rRNA taxonomic structure of the midgut microbiota of *M. festivus* samples (*n* = 167) at five black water and seven white water sites in Amazonia. Environmental vectors, based on five physicochemical characteristics measured at each site (*n* = 12), were fitted on the NMDS using the function "envfit" from *vegan* in R.

*Nyctotherus* sp. included an Halobacterota from the genus *Methanocorpusculaceae*, a potential endosymbiont of *Nyctotherus* sp.

## DISCUSSION

### *M. festivus* **midgut microbiota in both water types**

#### *Eukaryotic communities*

This is the first study investigating the Eukaryotic diversity within Amazonian wild fish gut using a molecular approach. Previous studies have focused on aquaculture fish, which were fed artificially and were reported to contain both a lower gut Eukaryotic diversity and parasitism rates than wild fish sampled in the present study (64, 65). For instance, we detected the presence of 135 non-host-related ASVs from 53 different

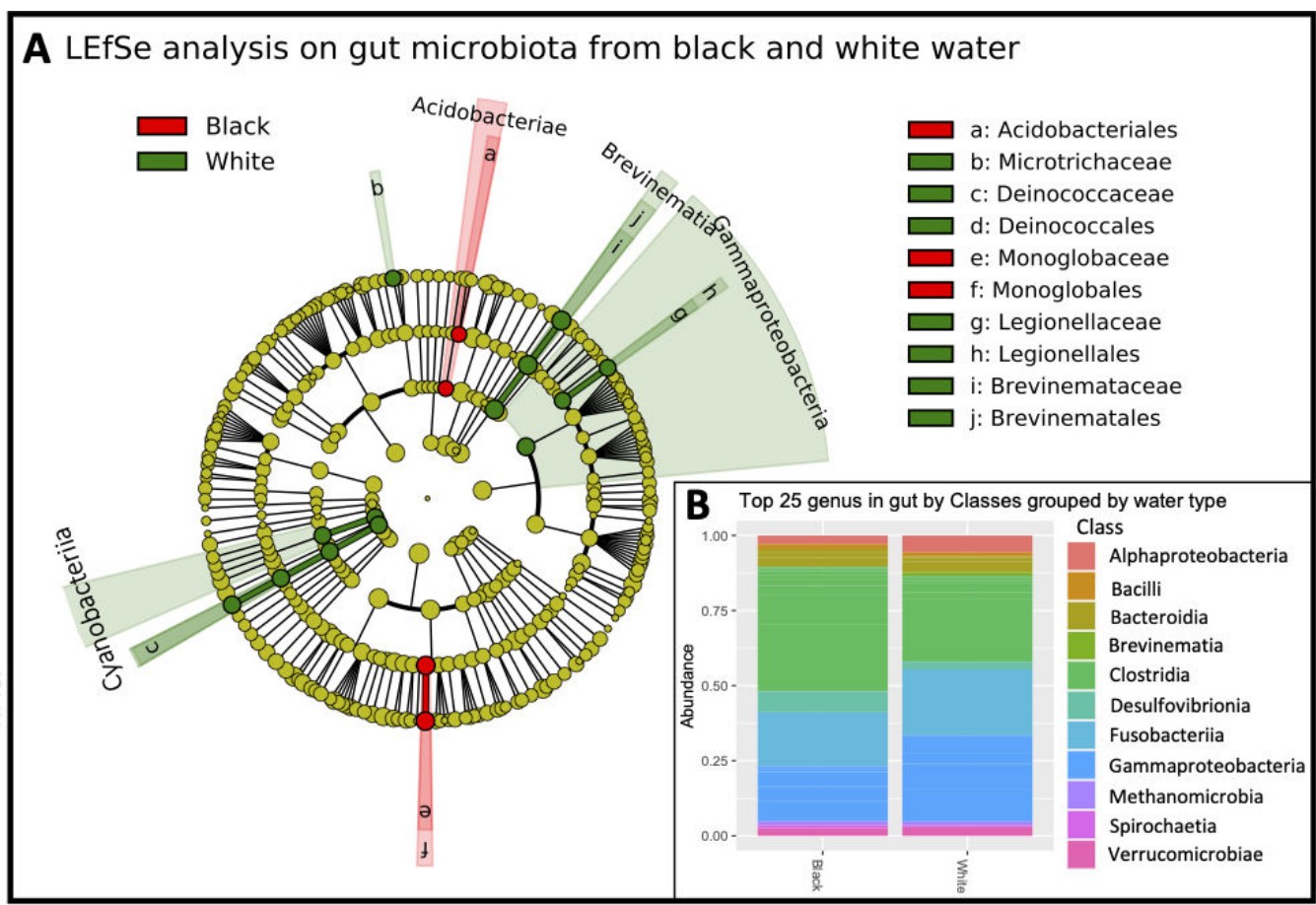

**FIG 5** (A) Linear discriminant analysis effect size based on the 16S rRNA taxonomic structure agglomerated by genus of the midgut microbiota of *M. festivus* (*n* = 167) sampled at five black water and seven white water sites. The water type at each site was considered as the discriminant variable. We detected 17 discriminant features considering a minimal LDA score of 3 and a *P*-value of $1 \times 10^{-3}$ for the Kruskal-Wallis test. (B) Relative abundance bar plots of the 25 most abundant genera detected in the same data set. Samples are grouped by water type, and each genus is colored according to its taxonomic class.

genera across our 167 *M. festivus* samples. Most of the observed Eukaryotic diversity in our samples was composed of food-related taxa (i.e., Arthropoda, Streptophyta, Rotifera, Chlorophyta, and Annelida) and potentially gut dwelling taxa (i.e., Ciliophora, Platyhelminthes, Nematoda, and Acanthocephala). Only nine ASVs were fungi, with Ascomycota being the most abundant phylum from this kingdom. Ascomycota were previously reported as the main fungal taxa within fish's gut samples (66, 67).

Ciliophora, Platyhelminthes, and Nematoda were the potentially gut dwelling phyla with the highest prevalence in *M. festivus* midgut samples (Fig. 2). For Ciliophora, the genus *Nyctotherus* sp. was detected in 53 samples. This genus has only been reported once in fish and was suggested to be an endosymbiont of fish (68). It has also been recorded as an intestinal parasite of roaches and in herbivorous reptiles (69). These ciliates are known for their important methane production caused by their endosymbiosis with a methanogenic Archaea (70). While the pathogenicity of *Nyctotherus* sp. is still to be demonstrated for fish (71), the high concentrations of methanotrophic bacteria in the fish host gut could lead to important chemical changes in infected animals, potentially resulting in a shift of microbial communities (72). For instance, *Nyctotherus* sp. infections in pet turtles have been reported to cause diarrhea, dehydration, weight loss, and passage of undigested food in the feces (73). By the same token, Platyhelminthes, as Cestoda and Trematoda, are known as common parasites in detritivore tropical fish (35, 74). Their distribution and prevalence are of particular interest since they cause important health problems for riverine communities (75). For nematodes in this study,

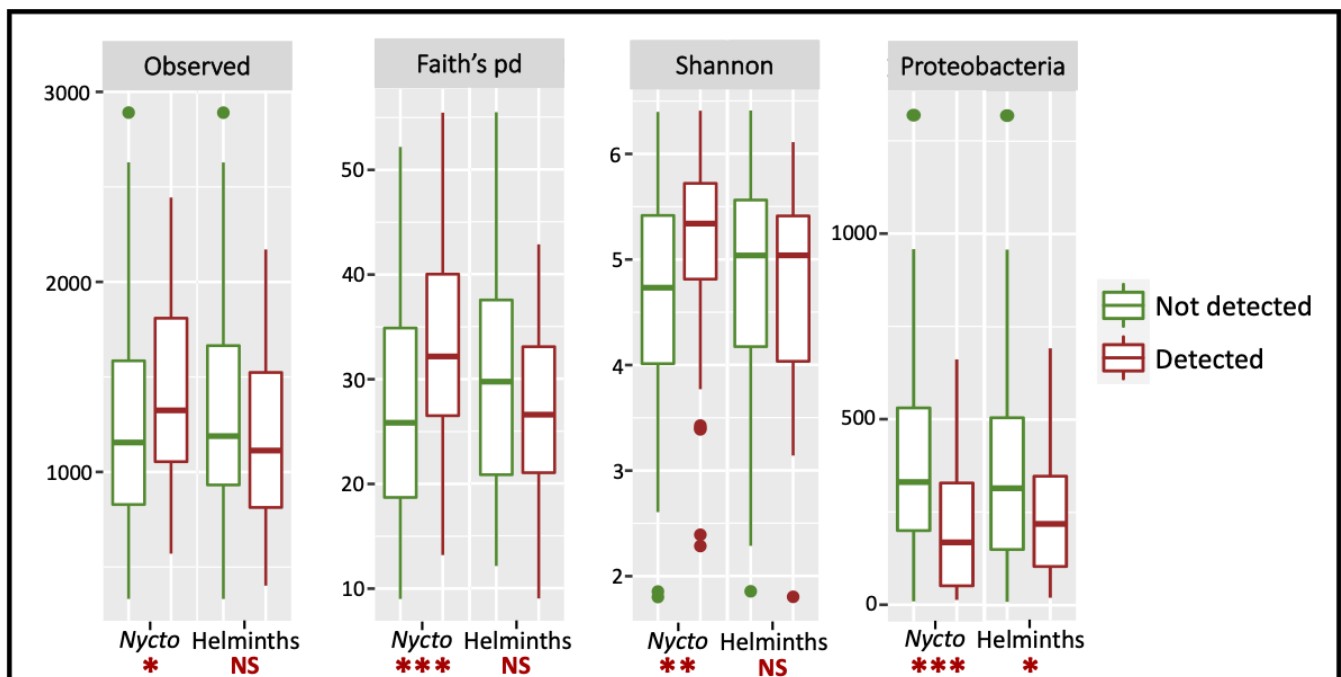

**FIG 6** Boxplot representing the 16S rRNA microbiota alpha diversity in midgut samples of *M. festivus* (*n* = 167) collected at 12 sites in central Amazonia. We measured the total number of ASVs observed (observed diversity), the Faith's phylogenetic diversity, the Shannon diversity, and the total number of ASVs from the phylum Proteobacteria in samples. Samples were clustered according to the detection of *Nyctotherus* sp. (*n* = 53) or helminths (*n* = 38) in fish midgut samples. We included the results of the two-sided two-sample Wilcoxon test comparing the alpha diversity of samples from which a parasite taxon was detected or not detected for each type of parasite (i.e., *Nyctotherus* sp. and helminths). NS stands for not significant.

most detections were of the genus *Spirocamallanus*, a genus of nematode frequently infecting Amazonian fish species, including cichlids, and causing important economic losses (37, 76).

Even though we detected several genera of known parasitic taxa that could infect *M. festivus*, our experimental design does not allow concluding about an actual infection from helminths (i.e., Platyhelminthes or Nematodes). DNA-based methods do not make a difference between living and dead specimens. Indeed, some Platyhelminthes or Nematodes could potentially originate from *M. festivus* feeding on a different life stage of the species, from a food source that was infected by the parasite, or even from free-living species closely related to a parasitic taxon. For instance, Camallanidae, an order of Nematode detected in samples, are known to infect copepods as a secondary host (77), and Tetraphyllidea, an order of Cestoda, is specifically infecting cartilaginous fish as a definite host (78). Furthermore, DNA-based methods do not make a difference between living and dead specimens, augmenting the risk of wrongfully concluding about an infection. Even though we only considered known gut dwelling taxa in our analyses, this pitfall needs to be considered in the interpretation of our results. Still, the ciliate *Nyctotherus* sp. can confidently be considered as a parasite as this gut dwelling genus is not known infecting food of *M. festivus* and could disperse by vertical transfer between hosts (69).

While the prevalence of nematodes and ciliates was independent to the water type, Platyhelminthes were more prevalent in *M. festivus* midgut samples collected in white water sites than in black water sites (Fig. 3). The mean prevalence of Platyhelminthes at black water sites was as low as 2%, compared to 26% at white water sites. Since our study is the first to compare the parasitism in black and white water environments, additional testing using an optimized framework for parasitism assessment in both water types is required to conclude with confidence about the higher Platyhelminthes parasitism rate in white water. Still, trematodes and cestodes have complex life cycles, which usually

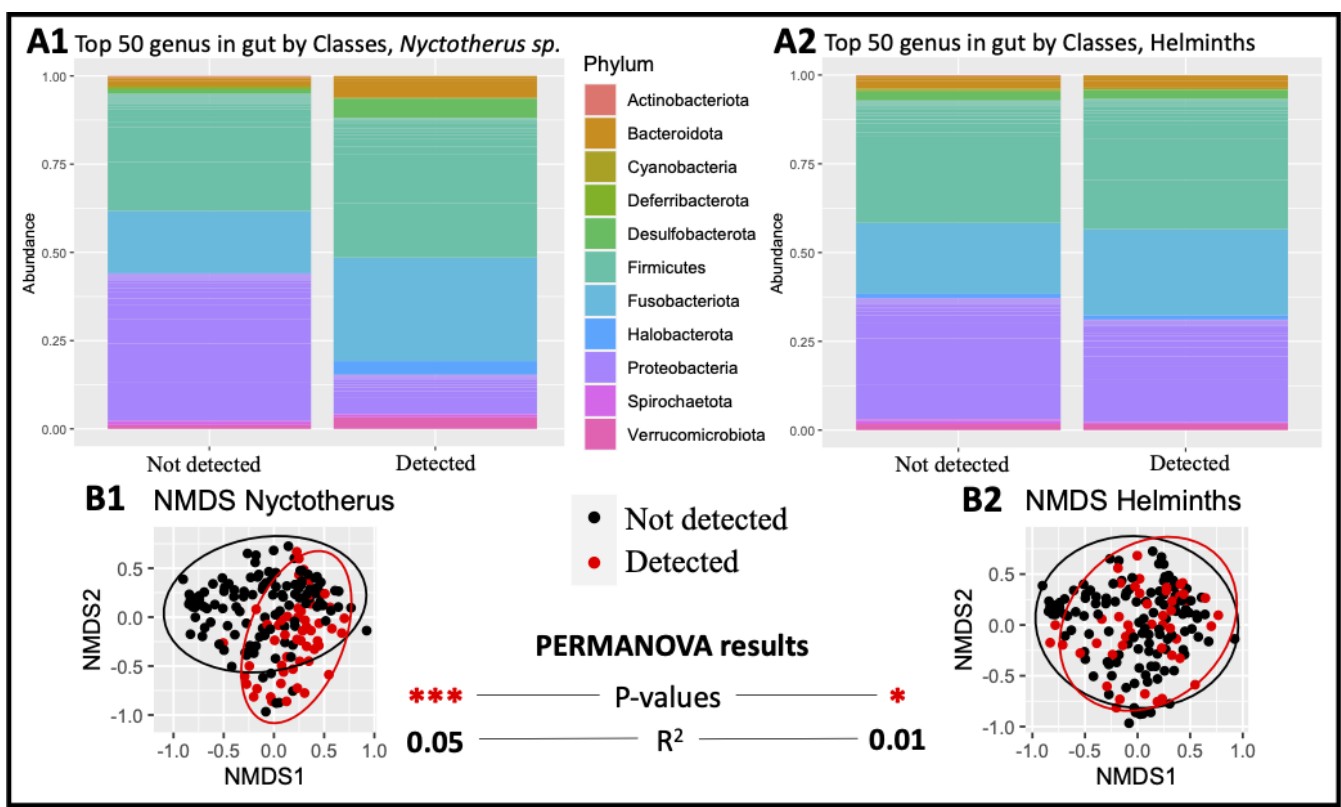

**FIG 7** (A) Relative abundance bar plots of the 50 most abundant genera detected in midgut samples of *M. festivus* (*n* = 167) clustered according to the detection of *Nyctotherus* sp. or helminths in fish midgut samples. (B) Non-metric multidimensional scaling plots based on the Bray-Curtis distances between samples of the 16S rRNA midgut microbiota of *M. festivus* (*n* = 167) at the ASV level. Samples in which *Nyctotherus* sp. or helminths have been detected were clustered together. Ninety-five percent confidence ellipse was included in the NDMS for each group. The results of the PERMANOVA based on four factors are also presented for both parasite types. The NMDS was produced with Vegan from R.

include the parasitism of multiple organisms and some life stages in the water column. Since black water has very restrictive physicochemical characteristics (18, 23) and tends to host a lower fish density than white water rivers (19), it would make sense that parasitic Platyhelminthes have difficulty thriving in black water. In the contrary, there was no dependency between the prevalence of Nematoda, *Nyctotherus* sp., and water type. Again, there is no point of comparison in the literature, but the simpler mechanisms of *Nyctotherus* sp. infection, mostly by vertical transmission, could reduce the influence of the environment on their infection rate (69). While these results are a good starting point, the addition of other studies on the matter would add points of comparison and help characterizing the parasitism rate in both water types.

Overall, the observed Eukaryotic diversity in our samples was 10-fold lower than the bacterial diversity observed in the same samples, considering we detected 532 different bacterial genera for 53 eukaryotic genera. In this sense, we observed a lower contribution of Eukaryotic taxa in the sympatric microfauna of the gut of *M. festivus*, with eukaryotic diversity dominated by pathogenic taxa in our samples. However, the low diversity observed could be partially caused by the differences between amplification methods. While we used blocking primers to reduce host DNA amplification, an important portion of our libraries was still composed of host DNA, potentially preventing the detection of multiple rare Eukaryotic taxa. Moreover, there could be a primer bias associated to 18S rRNA universal primers, favoring the detection of certain taxa (50, 79).

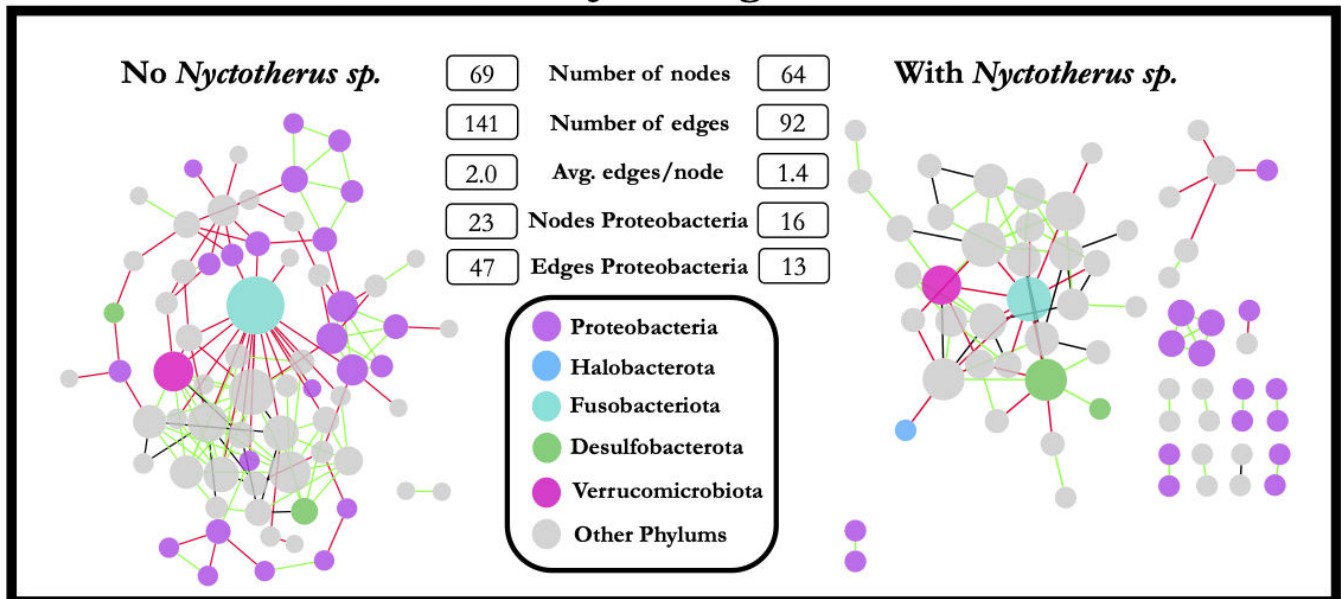

**FIG 8** Co-abundance networks based on the taxonomic structure of the midgut microbiota of *M. festivus* that are infected (*n* = 53) and not infected (*n* = 114) by *Nyctotherus* sp. The two networks were produced using the CoNet application in Cytoscape using the abundance of the 100 most abundant genera present in our 16S rRNA metabarcoding data set. Red edges (links) represent negative correlations between the abundance of two nodes (genus), while green edges represent positive correlations. We calculated the number of nodes, edges, the ratio of edges per node, the number of nodes from the phylum Proteobacteria, and the number of edges from the phylum Proteobacteria present in each network. The main taxonomic classes explaining the differences between the two networks are identified in the legend.

## Bacterial communities

The important physicochemical dissimilarity between black and white water environments had a limited influence on the midgut bacterial communities of *M. festivus*. For instance, alpha diversity was not significantly different when comparing the bacterial communities of gut samples from both water types. In addition, only the chlorophyll A concentration was significantly correlated to the microbiome beta diversity in the "Envfit" analysis. In the NMDS fitted with environmental vectors, the Bray-Curtis index from white water fish gut samples is aligned in the same axis as the chlorophyll A in the NMDS (Fig. 4). Fish living in a more productive environment could present different feeding habits, ultimately leading to changes in gut microbial communities (80). However, none of the four other physicochemical parameters differentiating black and white waters were significantly correlated to sample distribution in the graphical space. These results support a low influence of the restrictive environmental conditions of the black water environment (low ion concentration and acidic pH). On the contrary, environmental parameters associated to the alimentation of the host seem to have a higher influence. Since the gut microbiota of fish is relatively isolated from the water, changes in physicochemical characteristics could have a limited influence on its communities. This isolation from the environment could lead to a more important role of host-related factors, as alimentation and genotype, on structuring the gut microbiome (15, 41, 81).

The abundance of three important taxonomic classes (i.e., Gammaproteobateria, Brevinematia, and Clostridia) and two taxonomic classes which ecology is associable with specific characteristics of the different water types (i.e., Acidobacteriales and Cyanobacteria) could be the result of dispersal from the environment to the microbiota of *M. festivus*. The LEfSe analysis detected a higher abundance of Gammaproteobacteria and Brevinematia in the microbiota of *M. festivus* from white water sites and a higher

abundance of a family of Clostridia in samples from black water sites (Fig. 5A). These three taxonomic classes represent major components of the midgut microbiota of *M. festivus* since their relative abundance is high (Fig. 5B). In addition, high relative abundance of Gammaproteobacteria and Clostridia was, respectively, associated with the gut microbiome of fish from white and black water environments in a previous study (15). While these taxa could be recruited from the water directly, they could also be favored in a gut ecological niche that is specific to a given water type. For instance, the important environmental dissimilarity between both water types could lead to differences in the metabolism, feeding habits, or stress factors in fish living in different environments. These factors are known to influence on the taxonomic structure of fish gut microbiota (2). We also detected a higher relative abundance of Acidobacteriales in the microbiota of fish coming from black water sites. These acidophilic bacteria probably thrive in acidic black water. Similarly, we detected a higher abundance of Cyanobacteria in fish from white water, furthering the potential dispersal of bacteria associated with the higher chlorophyll A concentration in white water sites (Fig. 4). While these two taxonomic classes are not major components of the midgut microbiota of *M. festivus* (Fig. 5B), their presence could represent cases of water-to-fish dispersal. For both Acidobacteriales and Cyanobacteria, we can link the biology of the bacteria to typical characteristics of each water type. Dispersal of bacterioplankton taxa from water to fish gut was documented for discus fish (12), rainbow trout (82), and Atlantic Salmon (13). However, it may be hazardous to directly link these bacterial strains to the water bacterioplankton in this study because we did not conduct a time series of sampling to account for a possible decoupling between their respective taxonomic dynamics in water and fish gut.

In contrast, we detected a more important influence of the genetic population from which gut samples originated than their water type of origin. In the PERMANOVA, the bacterial communities of samples originating from related fish, fish from the same genetic population, tended to be more alike than the ones coming from unrelated fish collected from the same water type. This result supports a higher influence of genetic-related factors than environmental factors in shaping the taxonomic composition of *M. festivus* gut microbiome. These results concord with previous findings, which highlighted a higher influence of the phylogeographic relatedness between fish than ecological variations on structuring the gut microbiota of cichlids (83, 84). *M. festivus* is known for its important parental care investment, during which juveniles stay very close to both parents (43). This type of behavior could facilitate a vertical transfer of gut microbial communities. Furthermore, some genotypes that are specific to a given genetic population could have the ability to recruit specific bacteria, ultimately leading to a higher similarity of fish genetically linked. Here, *M. festivus* is a species of particular interest as some closely related individuals are known to originate from drastically divergent environmental conditions (i.e., from black and white water sites) (49). This characteristic of *M. festivus* helps deciphering the effect of the genotype and environmental conditions, two factors that are generally highly correlated in natural systems (85, 86). Naturally, further study of the gut microbiota of *M. festivus* in light of its genetics would help to disentangle the relative contribution of these two complex factors for cichlids. In future research, doing a controlled transplant experiment with genotyped fish from different environments and genotypes could highlight fish microbiome adaptations to a changing environment.

Overall, our results support a low but present effect of the environmental dissimilarity between black and white water environments on the taxonomic structure of the midgut microbiota in *M. festivus*. As supported by previous research on cichlids, the genetic relatedness between sites seems to play a more important role than the environmental similarity between sites on the taxonomic structure of the midgut microbiota of *M. festivus*.

## Influence of parasites on the gut microbiota

### Helminths

As the presence of Nematodes and Platyhelminths in gut samples had similar effects on the taxonomic structure of *M. festivus* gut microbiota, we pooled both types of parasites in these analyses. According to our results, there seems to be no direct relationship between *M. festivus* gut microbiome and helminths occurrence. While most past studies are congruent with our results, supporting a low influence of helminth infections on taxonomic structures of gut microbiota in both fish and mammals (87–89), another study detected significant interactions between the microbiota and helminth infections in zebrafish (90). Overall, it seems like controlled experiments are better at detecting the influence of helminths on the microbiota since they reduce the interindividual variability by controlling for host sex, genotype, and environment (91, 92). Furthermore, controlled experiment can ensure that the helminth is truly infecting the fish, while our experimental design can only detect its presence in the gut. Still, intestinal helminths constantly secrete products that could modulate the metabolism of microbial communities, and there is a clear gap in knowledge about potential host-parasite-microbiota interactions for these taxa.

### Nyctotherus sp.

Conversely, infection by *Nyctotherus* sp. had a major impact on *M. festivus* microbiota. First, the three alpha diversity metrics considered were significantly higher for gut samples infected by *Nyctotherus* sp. (Fig. 6). In addition, infections by *Nyctotherus* sp. explained five times more variance than the detection of helminths in the PERMANOVA based on Bray-Curtis index, translating into a clear spatial structuration of samples according to this factor in the PCoA (Fig. 7B). Unhealthy fish usually have a less diversified microbiota accompanied with the presence of pathology-specific bacterial taxa that should otherwise be absent (29, 93). Here, we documented a reduction in diversity (Fig. 6) and relative abundance of Proteobacteria in the microbiota, where they were replaced by higher relative abundances of other classes of bacteria (i.e., Bacteroidota, Desulfobacterota, Fusobacterota, Verrucomicrobiota, and Halobacterota) (Fig. 7A). Proteobacteria are a major component of the gut microbiota of bony fish (94, 95), and a reduction in their abundance and diversity associated with *Nyctotherus* sp. infection might result in a dysbiosis of the gut (17, 96). In addition, we detected high abundances of Halobacterota, a class of methanogenic Archaea absent from samples free of *Nyctotherus* sp., in samples infected by *Nyctotherus* sp. More specifically, the genus *Methanobrevibacter* is a known endosymbiont of multiple species of *Nyctotherus* sp. (70, 97). In addition, in fish infected by *Nyctotherus* sp., we observed an increased total diversity, mainly driven by an increased Halobacteria, Verucomicrobiales, and Eubacteriales diversity. While a clear definition of a healthy gut microbiota is still lacking for cichlids, studies tend to support that Fusobacteria, Proteobacteria, and Firmicutes should dominate the microbial communities in the gut, with the three previously reported to represent approximately 80% of the microbiota (2, 84). Thus, the high abundance of ciliates and its endosymbiont, which are both competing for resources with other taxa, may have major impacts on the gut microbial community structure of *M. festivus*.

Network analysis provides hallmarks of microbiota dysbiosis (13, 98–101). A complex and highly interconnected co-abundance network is representative of a more resilient and healthy microbiota, one where the removal of a genus of bacteria will have minimal impact on its structure due to numerous other edges solidifying each node (102). Likewise, the co-abundance network based on fish infected by *Nyctotherus* sp. had a reduced complexity and connectivity when compared to the network of fish without *Nyctotherus* sp., indicating a dysbiosis of the gut microbiota (Fig. 8). In fact, the co-abundance network based on fish with *Nyctotherus* sp. had 30% fewer edges (connections) per node when compared to the one based on fish without the ciliate. This major difference between the two experimental groups is mainly caused by the near absence

of Proteobacteria in the central network based on fish infected with *Nyctotherus* sp. Moreover, in fish without *Nyctotherus* sp., Proteobacteria are making a high number of connections with other bacteria genera, which goes a long way at forming a complex and solid co-abundance network. A lot of these Proteobacteria genera are absent or make isolated connections in the network based on fish infected by *Nyctotherus* sp., ultimately leading to a flimsier network (Fig. 8). In addition, we only observed the presence of Halobacterota in the co-abundance network based on fish infected with *Nyctotherus* sp., furthering its potential role in the microbiome restructuration observed for these fish. By causing a modification of the structure of the microbiota of its host, *Nyctotherus* sp. potentially reduces the metabolism, the immunity, and the survivability of infected fish (2, 103).

## Conclusion

In this study, we combined a dual 16S and 18S metagenomic approach, with *M. festivus* genotypes and environmental parameters at 12 sites to describe the interactions among the microbiota, parasite infections, and the environment. We observed a higher prevalence of gut dwelling Platyhelminthes in fish from white water than from those of black water sites, while prevalence rates for ciliates and nematodes were similar for fish from both water types. In addition, the phylogenetic relatedness between fish was shown to have twice as much impact on gut microbiota than the water type, suggesting a higher influence of the genotype in recruiting bacterial symbionts. Above all, we reported the first description of microbiota dysbiosis resulting from an infection by *Nyctotherus* sp., which presence was associated with a reduction in Proteobacteria diversity and high abundances of a methanogenic Archaea. Our study provides novel insights into the complex ecological interactions between fish, their gut microbiome, and their associated parasitic communities.

## ACKNOWLEDGMENTS

We gratefully acknowledge help from all members of the Derome and Val laboratory. Thank you to the IBIS Plateforme d'Analyse Génomique for the sequencing and to ICMBIO/Instituto Chico Mendes de Conservação da Biodiversidade for in situ support for fish collection and for issuing the permit to transport biological samples (permit number 29837/18).

This work was part of the ADAPTA project at INPA and was supported by the INCT ADAPTA (CNPq/FAPEAM) and INPA/MCTI grants to A.L.V., the Natural Sciences and Engineering Research Council of Canada (NSERC) Discovery grant (grant #6333) and the Canada-Brazil Awards Joint Research Project to N.D., and the NSERC BESC M Alexander-Graham-Bell, the FRQNT Master Scholarship grant, the MITACS GlobaLink grant, the Ressources Aquatiques Québec internship grant, and the André Darveau grant to N.L..

N.L., F.E.S., N.D., and A.V. designed the experiment. F.S., N.L., A.H., N.D., and A.V. organized sampling expeditions. F.S., N.L., A.H., and N.D. sampled fish during field expeditions. N.L., F.S., and A.H. processed samples in the laboratory (fish dissections and DNA extractions). N.L. performed bioinformatic analyses. N.L. wrote the manuscript. All authors reviewed the manuscript.

## AUTHOR AFFILIATIONS

[1]Department of Biology, Laval University, Quebec City, Quebec, Canada
[2]Institut de Biologie Intégrative et des Systèmes, Quebec City, Quebec, Canada
[3]Department of Environment and Genetics, Centre for Freshwater Ecosystems, Wodonga, Victoria, Australia
[4]Laboratory of Ecophysiology and Molecular Evolution, Brazilian National Institute for Research of the Amazon, Manaus, Brazil

## AUTHOR ORCIDs

Nicolas Leroux  http://orcid.org/0000-0003-1198-9435
Francois-Etienne Sylvain  http://orcid.org/0000-0002-1763-0434
Adalberto Luis Val  http://orcid.org/0000-0002-3823-3868
Nicolas Derome  http://orcid.org/0000-0002-2509-6104

## FUNDING

| Funder | Grant(s) | Author(s) |
| --- | --- | --- |
| Instituto Nacional de Ciência e Tecnologia Centro de Estudos das Adaptações da Biota Aquática da Amazônia (INCT-ADAPTA) | | Adalberto Luis Val |
| Gouvernement du Canada \| Natural Sciences and Engineering Research Council of Canada (NSERC) | 6333 | Nicolas Derome |
| Canada-Brazil Awards Joints Research Project | | Nicolas Derome |
| Gouvernement du Canada \| Natural Sciences and Engineering Research Council of Canada (NSERC) | | Nicolas Leroux |
| FRQ \| Fonds de recherche du Québec – Nature et technologies (FRQNT) | | Nicolas Leroux |
| Mitacs (Mitacs Canada) | | Nicolas Leroux |
| Ressources Aquatiques Québec | | Nicolas Leroux |
| André Darveau grant | | Nicolas Leroux |

## AUTHOR CONTRIBUTIONS

Francois-Etienne Sylvain, Conceptualization, Funding acquisition, Methodology, Resources, Writing – review and editing | Aleicia Holland, Methodology, Resources, Writing – review and editing | Adalberto Luis Val, Conceptualization, Methodology, Project administration, Resources, Supervision, Writing – review and editing | Nicolas Derome, Conceptualization, Methodology, Project administration, Supervision, Validation, Writing – review and editing.

## DATA AVAILABILITY

The scripts, amplification primer list, and data sets used for the statistical analysis of this project are freely available on the Open Science Framework (https://osf.io/znjq2/?view_only=00588956d5ef4f8b8fc3814c4838255a). Raw sequence reads and metadata are deposited in the SRA (BioProject accession number PRJNA997738).

## ETHICS APPROVAL

This study was carried out in accordance with the recommendations of the Ethics Committee for the Use of Animals of the Instituto Nacional de Pesquisas da Amazonia (INPA). The permit (number 29837–18 as of 23 March 2021) was approved by the Ethics Committee for the Use of Animals of INPA and the Animal Protection Committee of Laval University (Quebec, Canada) (permit number 2018021–1).

## ADDITIONAL FILES

The following material is available online.

Open Peer Review

**PEER REVIEW HISTORY (review-history.pdf).** An accounting of the reviewer comments and feedback.

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
