## [Reviewer comments · Microbiology Spectrum]

Microbiology Spectrum

Gut microbiota of an Amazonian fish in a heterogeneous riverscape: Integrating Genotype, Environment and Parasitic infections

Nicolas Leroux, Francois-Etienne Sylvain, Aleicia Holland, Adalberto Luis Val, and Nicolas Derome

Corresponding Author(s): Nicolas Leroux, Universite Laval

Review Timeline:

Submission Date:	July 20, 2022
Editorial Decision:	September 2, 2022
Revision Received:	June 15, 2023
Editorial Decision:	July 8, 2023
Revision Received:	July 14, 2023
Accepted:	July 14, 2023

Editor: John Chaston

Reviewer(s): Disclosure of reviewer identity is with reference to reviewer comments included in decision letter(s). The following individuals involved in review of your submission have agreed to reveal their identity: Mikhail Solovyev (Reviewer #1)

Transaction Report:

DOI: <https://doi.org/10.1128/spectrum.02755-22>

September 2, 2022

Mx. Nicolas Leroux
Laval University
Biology
1030 avenue de la Médecine
Quebec, QC
Canada

Re: Spectrum02755-22 (Gut microbiota of an Amazonian fish in a heterogeneous riverscape: Integrating Genotype, Environment and Parasitic infections)

Dear Mx. Nicolas Leroux:

Thank you for submitting your manuscript to Microbiology Spectrum. It has been reviewed by two experts, and both viewed the manuscript positively and had some suggestions for revision. I have therefore decided to request 'Modifications', which means that if you choose to, you can submit a revised version of the text following the details outlined below, and that it is likely that I will send the revision out for a second round of review.

You will see that each reviewer has suggested several places where you can clarify the methods or, if necessary, add additional analyses. I generally agree with their concerns. As you revise, I confirm to you that Spectrum's scope is that "rather than making subjective evaluations of potential impact, Microbiology Spectrum publishes research studies that are of high technical quality and are useful to the community." I will be supportive of revisions that meet this expectation, and encourage you to consider these criteria and explain them in your reviewer response during the revision.

Link Not Available

Sincerely,

John Chaston

Journals Department
Reviewer comments:

Reviewer #1 (Comments for the Author):

The paper entitled "Gut microbiota of an Amazonian fish in a heterogeneous riverscape: Integrating Genotype, Environment and Parasitic infections" focused on estimation of different factors that potentially affect on fish gut bacterial community structure. I very like that more and more studies started to include parasites as a potential factor that may modulate gut bacterial community. The present study is very well written, logically presented and statistical analyses were professionally conducted. In the same time, there are several main queries (from my point of view) that have to be addressed. Please see them below.

Lines 72-73. From these, four usually stand out as the most important: environmental conditions, host diet, genotype, and physiological condition

Note. In the introduction part the authors described environmental conditions, genotype, and physiological condition but nothing was mentioned about the effect of host diet on the bacterial community structure. I guess that such important factor as diet has to be mentioned as well.

Lines 175-176. Field trips were conducted from September to December 2018-2019 during the dry season.

Note. How did the authors estimate the potential effect of "year"? The microbiota of aquatic ecosystem can be altered from year to year. The mentioned period (dry season) included four months (September-December). Please clarify the potential effect of different months on the gut bacterial structure and support it by statistical analyses. For example, both March and May belong to "spring" season but various water physico-chemical characteristics could be significantly different between them.

Lines 195-197. A total of 167 midgut samples were collected and processed. DNA was extracted from 20 mg of a midgut segment for each fish. We specifically selected the middle section of the intestine to be constant in the tissues analysed.

Note. Please clarify the potential effect of content in midgut of fish on general bacterial structure of this gut segment. Did the authors estimate the midgut fullness? It is known that the diversity estimates in gut mucosa and gut content could be different. I guess that among 167 specimens there were fish with different levels of gut fullness.

Lines 458-462. For instance, we detected the presence of 135 nonhost-related ASVs from 53 different genera across our 167 *Mesonauta festivus* samples. In comparison, Minich et al. (2018) detected less than 50 Eukaryotic operational taxonomic units (OTUs) in their faeces and gut samples of aquaculture fish.

Note. Sorry, I do not understand why the authors try to find relations between the level of parasite infestation in fish populations from nature and fish cultivated under aquaculture conditions? It is expected that these communities will be very different.

Lines 487-501. Note. I very like that the authors discuss about the limitations of DNA-barcoding approach trying to explain observed results. For me, the most important limitation of this approach is impossibility to estimate the "size" of the factor "parasite infestation", I mean, such parameter as intensity level. Indeed, the lack of significant effect of parasite invasion in experiment where available information is categorized only as "yes" or "no" parasites in the fish gut is very vulnerable.

Lines 504-507 and 512-513. Note. I guess that the most important factor is the present or absent the first intermediated host for given cestode species rather than the type of water. Perhaps, in black water there are no specific zooplanktonic crustaceans that could be used as the first intermediate hosts by cestodes.

Lines 566-568. Similarly, we detected a higher abundance of Cyanobacteria in fish from white water, furthering the potential horizontal transfer of bacteria associated with the higher chlorophyll A concentration in white water sites.

Note. It is good example of horizontal transfer of bacterial taxa from aquatic compartments to fish gut. Similar observation was found by Parshukov et al. 2019.

Probably this ms (Supporting Information File S4) will be useful:

Variations of the intestinal gut microbiota of farmed rainbow trout, *Oncorhynchus mykiss* (Walbaum), depending on the infection status of the fish. 2019. *Journal of Applied Microbiology* 127(2) DOI: 10.1111/jam.14302

Lines 412 - 414. However, as these taxa were not detected in the water microbiome sampled in each water type, it was not possible to link these discriminant features to a putative horizontal transfer from the water microbiome. Lines 544-546. Since the gut microbiota of fish is relatively isolated from the water, changes in physicochemical characteristics could have a limited influence on its communities. Lines 572-573. However, we cannot directly link these bacterial strains to the water bacterioplankton as we did not observe these in our water microbiome assessment. Lines 596-598. Overall, our results support a low but present effect of the environmental dissimilarity between black and white water environments on the taxonomic structure of the mid gut microbiota in *M. festivus*.

Note. Due to the author have not data associated with the bacterial community structure of water and fish gut estimated during several time points, low level of relationships between bacterial communities of water and fish gut is expected. Both type of

bacterial communities (from water and fish gut) could be characterized by different time that is needed to change them. It could be expected that bacterial community in water is more sensitive and changed faster than the bacterial community in fish gut even if, as the authors pointed out, the gut bacterial community originated from the surrounding water. I understand that sample collection in nature is not the same and easy as in laboratory but the limited data from nature may led to erroneous conclusions.

Unfortunately, I did not find any speculations regarding to feeding features of *M. festivus* from black and white waters. The authors mentioned that this fish is detritivorous (line 157) and that host diet is very important factor (line 73) but there are no any analysis or speculations associated with potentially different microbiota of sediments itself in the bottom of white and black waters or differences in community of invertebrates inhabiting the bottom sediments of those places (both sediments and invertebrates may possess by specific microbiomes). That fact that the structure of community of invertebrates are different between black and white waters is partially supported by differences in the structure of fish parasites (as it was shown in the present ms) required invertebrates as the first intermediate hosts (for example, cestodes). Moreover, the authors may have such data because 18S rRNA analysis was conducted. It could be interesting to compare the gut content of fish from black and white water using 18 S data and, consequently, speculate about similarity of fish diets from studied different sites.

Probably these two papers about microbiota of fish parasites are also will be useful.

1) The effect of diet on the structure of gut bacterial community of sympatric pair of whitefishes (*Coregonus lavaretus*): One story more. 2019PeerJ 7(9) DOI: 10.7717/peerj.8005

2) Composition of the microbial communities in the gastrointestinal tract of perch (*Perca fluviatilis* L. 1758) and cestodes parasitizing the perch digestive tract. 2019. Journal of Fish Diseases 43(4). DOI:10.1111/jfd.13096

Reviewer #2 (Comments for the Author):

The authors present an interesting study of the gut microbiomes of *M. festivus*, an Amazonian wild fish, and the impact of genotype, water quality, and parasite incidence on the gut microbiome. The manuscript is well-written and then study and analyses are generally sound. That said, I do have several minor comments for the authors to consider:

Line 98: Recommend replacing "horizontal transfer", which is a term that invoked genetic mechanisms, with "dispersal", which invokes the intended ecological mechanism.

Line 145: I have trouble following the defined scenario here, because it is presented in the context of strong versus weak environmental effects. However, the expecations outlined as a result of these effects includes genetic components. It thus rather seems that the context of the spectrum being defined in this scenario is actually about the strength of the environmental effects relative to the strength of the genetic effects. If that's an accurate interpretation of this scenario, it would help readers to be explicit. If I'm mistaken, clarification in this passage would help.

Line 198: Does the DNeasy kit include mechanical disruption of the samples (e.g., bead beating)? If not, it would be worth articulating whether this could be a caveat impacting the results, as many taxa may be recalcitrant to other forms of cellular lysis.

Line 222: Please specify the read length and whether the data produced by the MiSeq run was paired end.

Line 281: Phylogenetic diversity metrics are used, but it is not clear how the phylogeny was assembled. Please provide these details in the Methods.

Line 323: It isn't clear what is meant by "pooling does not influence the analysis of the results in the discussion"

Line 390: The methods seem to indicate that PERMANOVA was used to evaluate the association between sample covariates and beta-diversity, yet here it seems that an ordination based correlation analysis was used. Can the authors clarify why this approach was used to address this analysis and not PERMANOVA, which tends to be more robust for these types of investigations since it is not dependent upon the oridation?

Line 399: Adonis is often found to be sensitive to the order in which the regression covariates are listed. Have the authors validated that reordering genotype and water type in their PERMANOVA regression model produces consistent results to those reported here?

Line 567: Are the authors confident in their Cyanobacteria annotations? Could these annotations alternatively result from the presence of chloroplast DNA in the fish diet?

Staff Comments:

Preparing Revision Guidelines

Please return the manuscript within 60 days; if you cannot complete the modification within this time period, please contact me. If you do not wish to modify the manuscript and prefer to submit it to another journal, please notify me of your decision immediately so that the manuscript may be formally withdrawn from consideration by Microbiology Spectrum.

The paper entitled “Gut microbiota of an Amazonian fish in a heterogeneous riverscape: Integrating Genotype, Environment and Parasitic infections” focused on estimation of different factors that potentially affect on fish gut bacterial community structure. I very like that more and more studies started to include parasites as a potential factor that may modulate gut bacterial community. The present study is very well written, logically presented and statistical analyses were professionally conducted. In the same time, there are several main queries (from my point of view) that have to be addressed. Please see them below.

Lines 72-73. From these, four usually stand out as the most important: environmental conditions, host diet, genotype, and physiological condition

Note. In the introduction part the authors described environmental conditions, genotype, and physiological condition but nothing was mentioned about the effect of host diet on the bacterial community structure. I guess that such important factor as diet has to be mentioned as well.

Lines 175-176. Field trips were conducted from September to December 2018–2019 during the dry season.

Note. How did the authors estimate the potential effect of “year”? The microbiota of aquatic ecosystem can be altered from year to year. The mentioned period (dry season) included four months (September-December). Please clarify the potential effect of different months on the gut bacterial structure and support it by statistical analyses. For example, both March and May belong to “spring” season but various water physico-chemical characteristics could be significantly different between them.

Lines 195-197. A total of 167 midgut samples were collected and processed. DNA was extracted from 20 mg of a midgut segment for each fish. We specifically selected the middle section of the intestine to be constant in the tissues analysed.

Note. Please clarify the potential effect of content in midgut of fish on general bacterial structure of this gut segment. Did the authors estimate the midgut fullness? It is known that the diversity estimates in gut mucosa and gut content could be different. I guess that among 167 specimens there were fish with different levels of gut fullness.

Lines 458-462. For instance, we detected the presence of 135 nonhost-related ASVs from 53 different genera across our 167 *Mesonauta festivus* samples. In comparison, Minich et al. (2018) detected less than 50 Eukaryotic operational taxonomic units (OTUs) in their faeces and gut samples of aquaculture fish.

Note. Sorry, I do not understand why the authors try to find relations between the level of parasite infestation in fish populations from nature and fish cultivated under aquaculture conditions? It is expected that these communities will be very different.

Lines 487-501. Note. I very like that the authors discuss about the limitations of DNA-barcoding approach trying to explain observed results. For me, the most important limitation of this approach is impossibility to estimate the “size” of the factor “parasite infestation”, I mean, such parameter as intensity level. Indeed, the lack of significant effect of parasite invasion in experiment where available information is categorized only as “yes” or “no” parasites in the fish gut is very vulnerable.

Lines 504-507 and 512-513. Note. I guess that the most important factor is the present or absent the first intermediated host for given cestode species rather than the type of water. Perhaps, in

black water there are no specific zooplanktonic crustaceans that could be used as the first intermediate hosts by cestodes.

Lines 566-568. Similarly, we detected a higher abundance of Cyanobacteria in fish from white water, furthering the potential horizontal transfer of bacteria associated with the higher chlorophyll A concentration in white water sites.

Note. It is good example of horizontal transfer of bacterial taxa from aquatic compartments to fish gut. Similar observation was found by Parshukov et al. 2019.

Probably this ms (Supporting Information File S4) will be useful:

Variations of the intestinal gut microbiota of farmed rainbow trout, *Oncorhynchus mykiss* (Walbaum), depending on the infection status of the fish. 2019. Journal of Applied Microbiology 127(2) DOI: 10.1111/jam.14302

Lines 412 - 414. However, as these taxa were not detected in the water microbiome sampled in each water type, it was not possible to link these discriminant features to a putative horizontal transfer from the water microbiome. Lines 544-546. Since the gut microbiota of fish is relatively isolated from the water, changes in physicochemical characteristics could have a limited influence on its communities. Lines 572-573. However, we cannot directly link these bacterial strains to the water bacterioplankton as we did not observe these in our water microbiome assessment. Lines 596-598. Overall, our results support a low but present effect of the environmental dissimilarity between black and white water environments on the taxonomic structure of the mid gut microbiota in *M. festivus*.

Note. Due to the author have not data associated with the bacterial community structure of water and fish gut estimated during several time points, low level of relationships between bacterial communities of water and fish gut is expected. Both type of bacterial communities (from water and fish gut) could be characterized by different time that is needed to change them. It could be expected that bacterial community in water is more sensitive and changed faster than the bacterial community in fish gut even if, as the authors pointed out, the gut bacterial community originated from the surrounding water. I understand that sample collection in nature is not the same and easy as in laboratory but the limited data from nature may led to erroneous conclusions.

Unfortunately, I did not find any speculations regarding to feeding features of *M. festivus* from black and white waters. The authors mentioned that this fish is detritivorous (line 157) and that host diet is very important factor (line 73) but there are no any analysis or speculations associated with potentially different microbiota of sediments itself in the bottom of white and black waters or differences in community of invertebrates inhabiting the bottom sediments of those places (both sediments and invertebrates may possess by specific microbiomes). That fact that the structure of community of invertebrates are different between black and white waters is partially supported by differences in the structure of fish parasites (as it was shown in the present ms) required invertebrates as the first intermediate hosts (for example, cestodes). Moreover, the authors may have such data because 18S rRNA analysis was conducted. It could be interesting to compare the gut content of fish from black and white water using 18 S data and, consequently, speculate about similarity of fish diets from studied different sites.

Probably these two papers about microbiota of fish parasites are also will be useful.

- 1) The effect of diet on the structure of gut bacterial community of sympatric pair of whitefishes (*Coregonus lavaretus*): One story more. 2019PeerJ 7(9) DOI: 10.7717/peerj.8005
- 2) Composition of the microbial communities in the gastrointestinal tract of perch (*Perca*

fluviatilis L. 1758) and cestodes parasitizing the perch digestive tract. 2019. Journal of Fish Diseases 43(4). DOI:10.1111/jfd.13096

September 2, 2022
Mx. Nicolas Leroux
Laval University
Biology
1030 avenue de la Médecine
Quebec, QC
Canada

Re: Spectrum02755-22 (Gut microbiota of an Amazonian fish in a heterogeneous riverscape: Integrating Genotype, Environment and Parasitic infections)

Dear Mx. Nicolas Leroux:

Thank you for submitting your manuscript to Microbiology Spectrum. It has been reviewed by two experts, and both viewed the manuscript positively and had some suggestions for revision. I have therefore decided to request 'Modifications', which means that if you choose to, you can submit a revised version of the text following the details outlined below, and that It is likely that I will send the revision out for a second round of review.

You will see that each reviewer has suggested several places where you can clarify the methods or, if necessary, add additional analyses. I generally agree with their concerns. As you revise, I confirm to you that Spectrum's scope is that "rather than making subjective evaluations of potential impact, Microbiology Spectrum publishes research studies that are of high technical quality and are useful to the community." I will be supportive of revisions that meet this expectation, and encourage you to consider these criteria and explain them in your reviewer response during the revision.

<https://spectrum.msubmit.net/cgi-bin/main.plex?el=A7QF1CBMh5A5FRxj3F2A9fdjg991zN7RbO4ZYBFfBH0AZ>

ASM policy requires that data be available to the public upon online posting of the article, so please verify all links to sequence records, if present, and make sure that each number retrieves the full record of the data. If a new accession number is not linked or a link is broken, provide production staff with the correct URL for the record. If the accession numbers for new data are

not publicly accessible before the expected online posting of the article, publication of your article may be delayed; please contact the ASM production staff immediately with the expected release date.

Sincerely,

John Chaston
Editor, Microbiology Spectrum

Journals Department
Reviewer comments:

Reviewer #1 (Comments for the Author):

The paper entitled "Gut microbiota of an Amazonian fish in a heterogeneous riverscape: Integrating Genotype, Environment and Parasitic infections" focused on estimation of different factors that potentially affect on fish gut bacterial community structure. I very like that more and more studies started to include parasites as a potential factor that may modulate gut bacterial community. The present study is very well written, logically presented and statistical analyses were professionally conducted. In the same time, there are several main queries (from my point of view) that have to be addressed. Please see them below.

Lines 72-73. From these, four usually stand out as the most important: environmental conditions, host diet, genotype, and physiological condition.

Comment 1: In the introduction part the authors described environmental conditions, genotype, and physiological condition but nothing was mentioned about the effect of host diet on the bacterial community structure. I guess that such important factor as diet has to be mentioned as well.

Response: Diet was explicitly mentioned in the two previous sentences, lines 69-71: "The taxonomic composition and functional repertoire of the gut microbiota are determined by a wide range of factors. From these, four usually stand out as the most important: environmental conditions, host diet, genotype, and physiological condition".

Lines 175-176. Field trips were conducted from September to December 2018-2019 during the dry season.

Comment 2: How did the authors estimate the potential effect of "year"? The microbiota of aquatic ecosystem can be altered from year to year. The mentioned period (dry season) included four months (September-December). Please clarify the potential effect of different months on the gut bacterial structure and support it by statistical analyses. For example, both March and May belong to "spring" season but various water physico-chemical characteristics could be significantly different between them.

Response: The sampling was planned to minimize any bias due to temporal effect on microbial communities. First, fish and water collection were completed in two months (October, November). Second, both white (WW) and black water (BW) samples were collected the same month in both 2018 and 2019: October 2018: Rio Negro – Anavilhanas (BW), Lago Janauari (WW), Lago Catalão (WW), November 2018: Rio Negro - Barcelos (BW), Lago Janauaca (WW), Rio Manacapuru (WW); October 2019: Rio Branco (WW),; Lago do cemeterio (BW), Rio Negro - Santo Alberto (BW); November 2019: Lago des pirates (WW), Lago Téfê-Solimões (WW), and Lago Téfê (BW). Therefore, we are confident that temporal bias was minimal in differentiating black and white water bacterial and eukaryotic communities. Finally, the potential bias due to temporal effect was assessed by incorporating “Sampling time” in the permanova, which did not detect any effect on microbiota composition.

Lines 195-197. A total of 167 midgut samples were collected and processed. DNA was extracted from 20 mg of a midgut segment for each fish. We specifically selected the middle section of the intestine to be constant in the tissues analysed.

Comment 3: Please clarify the potential effect of content in midgut of fish on general bacterial structure of this gut segment. Did the authors estimate the midgut fullness? It is known that the diversity estimates in gut mucosa and gut content could be different. I guess that among 167 specimens there were fish with different levels of gut fullness.

Response: *M. festivus* is a herbivorous/detritivorous fish which mostly feed on periphyton a resource that is very abundant in Amazonian water in both wet and dry season. For this reason, most fish had their gut full. For this reason, we sliced the same midgut section in each fish including gut content. Including the gut content in the analysis was also favorable for detecting parasitic taxa and describing fish' diet. We did not consider gut fullness in the analysis and considered it a random factor affecting every site at the same rate.

Lines 458-462. For instance, we detected the presence of 135 nonhost-related ASVs from 53 different genera across our 167 *Mesonauta festivus* samples. In comparison, Minich et al. (2018) detected less than 50 Eukaryotic operational taxonomic units (OTUs) in their faeces and gut samples of aquaculture fish.

Comment 4: Sorry, I do not understand why the authors try to find relations between the level of parasite infestation in fish populations from nature and fish cultivated under aquaculture conditions? It is expected that these communities will be very different.

Response: Reviewer is right. The comparable studies are scarce and often use different primers or markers, target other organisms and different environments. We decided to remove this sentence.

Lines 487-501. Note. I very like that the authors discuss about the limitations of DNA-barcoding approach trying to explain observed results. For me, the most important limitation of this approach is impossibility to estimate the "size" of the factor "parasite infestation", I mean, such parameter as intensity level. Indeed, the lack of significant effect of parasite invasion in experiment where available information is categorized only as "yes" or "no" parasites in the fish gut is very vulnerable.

Lines 504-507 and 512-513.

Comment 5: I guess that the most important factor is the present or absent the first intermediated host for given cestode species rather than the type of water. Perhaps, in black water there are no specific zooplanktonic crustations that could be used as the first intermediate hosts by cestodes.

Response: This is a very interesting comment. According to the study of Nakajiima et al. (2017), Rio Negro black water mean density and biomass of mesozooplankton including potential intermediate copepod hosts of cestode were higher compared to the Rio Solimoes white water, by 2.8 and 2.0 times, respectively. Therefore, such results would not support the hypothesis according to which the low prevalence of cestode intermediate copepod host would explain the lower cestode prevalence in fish. However, the taxonomic information provided in Nakajiima et al. (2017) is too limited to state whether water color has an impact on intermediated host for any given cestode species.

Nakajima R, Rimachi EV, Santos-Silva EN, Calixto LSF, Leite RG, Khen A, Yamane T, Mazeroll AI, Inuma JC, Utumi EYK, Tanaka A. The density and biomass of mesozooplankton and ichthyoplankton in the Negro and the Amazon Rivers during the rainy season: the ecological importance of the confluence boundary. PeerJ. 2017 May 11;5:e3308. doi: 10.7717/peerj.3308. PMID: 28507821; PMCID: PMC5429737.

Lines 556-558. Similarly, we detected a higher abundance of Cyanobacteria in fish from white water, furthering the potential horizontal transfer of bacteria associated with the higher chlorophyll A concentration in white water sites.

Comment 6: It is good example of horizontal transfer of bacterial taxa from aquatic compartments to fish gut. Similar observation was found by Parshukov et al. 2019. Probably this ms (Supporting Information File S4) will be useful: Variations of the intestinal gut microbiota of farmed rainbow trout, *Oncorhynchus mykiss* (Walbaum), depending on the infection status of the fish. 2019. Journal of Applied Microbiology 127(2) DOI: 10.1111/jam.14302.

Response: We thank the reviewer for putting this publication to our attention. We added this sentence (Lines 564-566): “*Dispersal of bacterioplankton taxa from water to fish gut were documented for discus fish (Sylvain et al. 2017), rainbow trout (Parshukov et al. 2019) and Atlantic Salmon (Lavoie et al. 2021).*”.

Lines 412 - 414. However, as these taxa were not detected in the water microbiome sampled in each water type, it was not possible to link these discriminant features to a putative horizontal transfer from the water microbiome. Lines 544-546. Since the gut microbiota of fish is relatively isolated from the water, changes in physicochemical characteristics could have a limited influence on its communities. Lines 572-573. However, we cannot directly link these bacterial strains to the water bacterioplankton as we did not observe these in our water microbiome assessment. Lines 596-598. Overall, our results support a low but present effect of the environmental dissimilarity between black and white water environments on the taxonomic structure of the mid gut microbiota in *M. festivus*.

Comment 7: Due to the author have not data associated with the bacterial community structure of water and fish gut estimated during several time points, low level of relationships between bacterial communities of water and fish gut is expected. Both type of bacterial communities (from water and fish gut) could be characterized by different time that is needed to change them. It could be expected that bacterial community in water is more sensitive and changed faster than the bacterial community in fish gut even if, as the authors pointed out, the gut bacterial community originated from the surrounding water. I understand that sample collection in nature is not the same and easy as in laboratory but the limited data from nature may led to erroneous conclusions.

Response: The reviewer raises another interesting point here. We modified the sentence accordingly (Lines 567-574): “However, it may be hazardous to directly link these bacterial strains to the water bacterioplankton in this study because we did not conduct a time series of sampling to account for a possible decoupling between their respective taxonomic dynamics in water and fish gut.”.

Unfortunately, I did not find any speculations regarding to feeding features of *M. festivus* from black and white waters. The authors mentioned that this fish is detritivorous (line 157) and that host diet is very important factor (line 73) but there are no any analysis or speculations associated with potentially different microbiota of sediments itself in the bottom of white and black waters or differences in community of invertebrates inhabiting the bottom sediments of those places (both sediments and invertebrates may possess by specific microbiomes). That fact that the structure of community of invertebrates are different between black and white waters is partially supported by differences in the structure of fish parasites (as it was shown in the present ms) required invertebrates as the first intermediate hosts (for example, cestodes). Moreover, the authors may have such data because 18S rRNA analysis was conducted.

Comment 8: It could be interesting to compare the gut content of fish from black and white water using 18 S data and, consequently, speculate about similarity of fish diets from studied different sites.

Response: This is a very interesting comment. Here is the reason why we did not add this analysis in the previous version of the manuscript. There are only 18S ASVs potentially representing prey (rotifera, annelidae, arthropoda): 212, 225, 287, 420, 439, 496, 501, 523, 545, 559, 577, 585, 598, 611, 613, 618, 641, 652, 655, 656, 668, 673, 674, 717, 741, 764. However, only 13 gut samples contain at least one read from one of these 26 ASVs. Because these 26 ASVs are detected in a very small fraction of the dataset, we do not think that it is a suitable dataset for testing watercolor effect on diet.

Furthermore, when considering these 13 samples to perform a PERMANOVAs analysis, there were no significant differences between both water types.

```

Permutation test for adonis under reduced model
Terms added sequentially (first to last)
Permutation: free
Number of permutations: 10000

adonis2(formula = (tax_glom(Pruned_18S, taxrank = "genus", NArm = T)@otu_table) ~ Water_type * Genotype, data = data.frame(Pruned_18S@sam_data),
permutations = 10000, method = "bray")

```

	Df	SumOfSqs	R2	F	Pr(>F)
Water_type	1	0.4390	0.07458	0.9051	0.61474
Genotype	3	1.6580	0.28166	1.1395	0.08709
Water_type:Genotype	1	0.3945	0.06701	0.8133	0.93571
Residual	7	3.3951	0.57675		
Total	12	5.8865	1.00000		

```

---
Signif. codes:  0 '****' 0.001 '**' 0.01 '*' 0.05 '.' 0.1 ' ' 1

```

Probably these two papers about microbiota of fish parasites will also be useful.

The effect of diet on the structure of gut bacterial community of sympatric pair of whitefishes (*Coregonus lavaretus*): One story more. 2019PeerJ 7(9) DOI: 10.7717/peerj.8005
 2) Composition of the microbial communities in the gastrointestinal tract of perch (*Perca fluviatilis* L. 1758) and cestodes parasitizing the perch digestive tract. 2019. Journal of Fish Diseases 43(4). DOI:10.1111/jfd.13096

Reviewer #2 (Comments for the Author):

The authors present an interesting study of the gut microbiomes of *M. festivus*, an Amazonian wild fish, and the impact of genotype, water quality, and parasite incidence on the gut microbiome. The manuscript is well-written and then study and analyses are generally sound. That said, I do have several minor comments for the authors to consider:

Comment 1: Line 98: Recommend replacing "horizontal transfer", which is a term that invoked genetic mechanisms, with "dispersal", which invokes the intended ecological mechanism.

Response: The reviewer is right. "horizontal transfer" was replaced with the terms "dispersal" or "recruitment". Lines 95-98: *"Since the surrounding water may be a source of recruitment of environmental microbial strains to the fish gut microbiota (Sylvain et al. 2019, Lavoie et al. 2021), we posed the hypothesis that certain taxa of M. festivus' gut microbiome would be related to the water type (black or white)."*; Lines 543-544: *"could be the result of dispersal from the environment to the microbiota of M. festivus."*

Comment 2: Line 145: I have trouble following the defined scenario here, because it is presented in the context of strong versus weak environmental effects. However, the expectations outlined as a result of these effects includes genetic components. It thus rather seems that the context of the spectrum being defined in this scenario is actually about the strength of the environmental effects relative to the strength of the genetic effects. If that's an accurate interpretation of this scenario, it would help readers to be explicit. If I'm mistaken, clarification in this passage would help.

Response 2: Reviewer is absolutely right. The paragraph was rewritten accordingly. See lines 141-148: *"In any case, the interplay between genetic and environmental effect on gut microbiota taxonomic structure ranges between the following extreme scenario: strong environmental effects relative to weak genetic effects and vice versa. In the first case, fish should display similar microbial community shifts at similar environmental shifts (e.g. independent black water sites versus their respective connected white water sites). On the opposite, in the presence of a weak environmental effect and a strong genetic effect, fish from a same genetic population, but inhabiting contrasting environments, are expected to converge in terms of microbiota composition."*

Comment 3: Line 198: Does the DNeasy kit include mechanical disruption of the samples (e.g., bead beating)? If not, it would be worth articulating whether this could be a caveat impacting the results, as many taxa may be recalcitrant to other forms of cellular lysis.

Response: We did not use mechanical disruption of the samples by bead beating. Instead, samples were cut into small pieces and digested with lysis buffer. We agree that this may have limited the completeness of detection of microbial and parasitic taxonomic diversity. However, this bias remains the same for each sample, which should not affect comparisons between groups. We have included a sentence in the methods to emphasize this point. See lines 192 " *The middle section of the intestine was specifically selected and samples were cut into small pieces using sterile blades and digested with lysis buffer overnight. Although an additional step of*

mechanical disruption would have improved the detection of both microbial and parasitic taxonomic diversity, this potential bias should not affect comparisons between experimental groups. "

Comment 3: Line 222: Please specify the read length and whether the data produced by the MiSeq run was paired end.

Response: This information was provided. See lines 231-236: *"The demultiplexed fastq sequence files were processed through DADA2 (Callahan et al. 2016) using the function "ytfilterAndTrim" with the following parameters: two as the phred score threshold for total read removal, a maximum expected error of two for forward reads and four for reverse reads, a truncation length of 280 base pairs for forward reads and 200 base pairs for reverse reads for 16S rRNA library, and a length of 275 base pairs for forward reads and 250 base pairs for reverse reads for 18S rRNA library."*

This is a pair end sequencing for both 16S and 18S. This is now clarified line 221: *"Multiplex paired-end sequencing was performed using the MiSeq platform"*. We omitted the function *"mergePairs"* for read merging. This is now corrected lines 236-240: *"The filtered reads were then fed to the error rate learning, dereplication, merging and Amplicon Sequence Variant (ASV) inference steps using the functions "learnErrors", "derepFastq" mergePairs and "dada". Chimeric sequences were removed using the "removeBimeraDenovo" function with the "pseudo" method parameter."*

Comment 4: Line 281: Phylogenetic diversity metrics are used, but it is not clear how the phylogeny was assembled. Please provide these details in the Methods.

Response: Phylogenetic diversity was computed with Phyloseq using `estimate_pd(phylo)` function. The tree was computed with the simple agglomerative (bottom-up) hierarchical clustering method (UPGMA). Chao1 and Shannon were computed using `estimate_richness` function. Lines 282-286: *"Using the 16S rRNA Phyloseq object previously produced, we estimated alpha diversity indexes such as the observed number of ASVs (Chao1), and Shannon entropy (Shannon 1949) using estimate_richness function. Then, phylogenetic diversity (Faith 1992) was computed with using estimate_pd(phylo) function. The tree was computed with the simple agglomerative (bottom-up) hierarchical clustering method (UPGMA)."*

Comment 5: Line 323: It isn't clear what is meant by "pooling does not influence the analysis of the results in the discussion".

Response: We clarified the sentence line 328-329 as: *"We previously analysed independently the impact of each type of helminth detected and their pooling did not have any impact on the general significance of the results."*

Comment 6: Line 390: The methods seem to indicate that PERMANOVA was used to evaluate the association between sample covariates and beta-diversity, yet here it seems that an ordination based correlation analysis was used. Can the authors clarify why this approach was used to

address this analysis and not PERMANOVA, which tends to be more robust for these types of investigations since it is not dependent upon the ordination?

Response: PERMANOVA was used to evaluate associations between covariate genotype and water types and the gut microbiome. However, we opted for an ordination-based method for the comparison of the 5 environmental parameters. We have now separated the two paragraphs in the methods to better separate both analyses.

We selected an ordination-based method as it allowed us to share the environmental vectors and the data together as a graphical output highlighting the relative influence of each parameter on the gut microbiome structure. Considering your comment, we decided to run the proposed analysis on a PERMANOVA to make sure we did not miss any important results. Here are the results:

Comparison of both results; Envfit : “ chlorophyll a concentration in water: $R^2 = 0.084$, p-value = 0.001; DOC: $R^2 = 0.003$, p-value = 0.76; Silicate: $R^2 = 0.027$, p-value = 0.11; Conductivity: $R^2 = 0.027$, p-value = 0.11; pH: $R^2 = 0.024$, p-value = 0.14).

PERMANOVA :

```
Permutation test for adonis under reduced model
Terms added sequentially (first to last)
Permutation: free
Number of permutations: 50000

adonis2(formula = (tax_glom(A_16S_Phyloseq, taxrank = "Genus", NArm = T)@otu_table) ~ DOC + Silicate + Chl_a + Conductivity + pH, data =
data.frame(A_16S_Phyloseq@sam_data), permutations = 50000, method = "bray")

```

	Df	SumOfSqs	R2	F	Pr(>F)
DOC	1	0.936	0.01750	3.1197	0.00194 **
Silicate	1	0.701	0.01312	2.3381	0.01574 *
Chl_a	1	0.919	0.01719	3.0636	0.00252 **
Conductivity	1	1.238	0.02315	4.1272	0.00024 ***
pH	1	1.384	0.02589	4.6152	8e-05 ***
Residual	161	48.282	0.90316		
Total	166	53.459	1.00000		

```
---
Signif. codes:  0 '***' 0.001 '**' 0.01 '*' 0.05 '.' 0.1 ' ' 1
```

Results are very different in the PERMANOVA as they show a significant association between each environmental parameter and microbiome beta diversity. However, this is probably due to the very high number of comparisons performed by the PERMANOVA compared to the lower number of comparisons used in envfit which only considers environmental parameters and bray Curtis distances given to the function. Indeed, since this type of analysis cannot be easily plotted and shows all significant results, it is very difficult to highlight a pattern in the data. For these reasons, we think that the envfit analysis, while potentially less powerful in detecting significant interactions, leads to more interesting and interpretable results.

Comment 7: Line 399: Adonis is often found to be sensitive to the order in which the regression covariates are listed. Have the authors validated that reordering genotype and water type in their PERMANOVA regression model produces consistent results to those reported here?

Response: Yes, we previously did the PERMANOVA in both orders to consider this potential bias and, while results are not perfectly identical, they show the same patterns, as exemplified below with one of our main PERMANOVA ran at a low depth.

```

Permutation test for adonis under reduced model
Terms added sequentially (first to last)
Permutation: free
Number of permutations: 20000

adonis2(formula = A_16S_Phyloseq@otu_table ~ Water_type * Genotype, data = data.frame(A_16S_Phyloseq@sam_data), permutations = 20000, method = "bray")
      Df SumOfSqs      R2      F Pr(>F)
Water_type      1    1.376 0.02031 3.5052 5e-05 ***
Genotype         3    2.601 0.03839 2.2088 5e-05 ***
Water_type:Genotype 1    0.571 0.00843 1.4555 0.0575 .
Residual        161   63.205 0.93286
Total           166   67.754 1.00000
---
Permutation test for adonis under reduced model
Terms added sequentially (first to last)
Permutation: free
Number of permutations: 20000

adonis2(formula = A_16S_Phyloseq@otu_table ~ Genotype * Water_type, data = data.frame(A_16S_Phyloseq@sam_data), permutations = 20000, method = "bray")
      Df SumOfSqs      R2      F Pr(>F)
Genotype      3    2.963 0.04374 2.5161 5e-05 ***
Water_type     1    1.014 0.01497 2.5832 0.00050 ***
Genotype:Water_type 1    0.571 0.00843 1.4555 0.06025 .
Residual      161   63.205 0.93286
Total         166   67.754 1.00000
---
Signif. codes:  0 '****' 0.001 '***' 0.01 '**' 0.05 '.' 0.1 ' ' 1

```

Comment 8: Line 567: Are the authors confident in their Cyanobacteria annotations? Could these annotations alternatively result from the presence of chloroplast DNA in the fish diet?

Response: Numerous plants were detected in the gut using 18S marker. However, given that divergence time between Cyanobacteria and the oldest vascular plant fossil (tracheophytes) is about 460 MYA (Kenrick and Crane, 1997), it is very unlikely that Cyanobacteria annotation was corrupted with chloroplast sequences, using E-values threshold as low as 10×10^{-30} .

July 8, 2023

Mx. Nicolas Leroux
Universite Laval
Biology
1030 avenue de la Médecine
Quebec, QC
Canada

Re: Spectrum02755-22R1 (Gut microbiota of an Amazonian fish in a heterogeneous riverscape: Integrating Genotype, Environment and Parasitic infections)

Dear Mx. Nicolas Leroux:

Thank you for submitting your manuscript to Microbiology Spectrum. As you will see your paper is very close to acceptance. Please modify the manuscript along the lines I have recommended. As these revisions are quite minor, I expect that you should be able to turn in the revised paper in less than 30 days, if not sooner. If your manuscript was reviewed, you will find the reviewers' comments below.

I ask that you please revise the manuscript text to include your responses to Reviewer 2 at lines 390 and 399. I think these are acceptable responses, and as it is likely that peers will have similar questions, incorporating these responses into the text will improve the transparency and robustness of your arguments.

When submitting the revised version of your paper, please provide (1) point-by-point responses to the issues raised by the reviewers as file type "Response to Reviewers," not in your cover letter, and (2) a PDF file that indicates the changes from the original submission (by highlighting or underlining the changes) as file type "Marked Up Manuscript - For Review Only". Please use this link to submit your revised manuscript. Detailed instructions on submitting your revised paper are below.

Link Not Available

Sincerely,

John Chaston

Reviewer comments:

Reviewer #1 (Comments for the Author):

Thank you for your complete replies to my comments.

Preparing Revision Guidelines

To submit your modified manuscript, log onto the eJP submission site at <https://spectrum.msubmit.net/cgi-bin/main.plex>. Go to Author Tasks and click the appropriate manuscript title to begin the revision process. The information that you entered when you first submitted the paper will be displayed. Please update the information as necessary. Here are a few examples of required

updates that authors must address:

Please return the manuscript within 60 days; if you cannot complete the modification within this time period, please contact me. If you do not wish to modify the manuscript and prefer to submit it to another journal, please notify me of your decision immediately so that the manuscript may be formally withdrawn from consideration by Microbiology Spectrum.

September 2, 2022
Mx. Nicolas Leroux
Laval University
Biology
1030 avenue de la Médecine
Quebec, QC
Canada

Re: Spectrum02755-22 (Gut microbiota of an Amazonian fish in a heterogeneous riverscape: Integrating Genotype, Environment and Parasitic infections)

Dear Mx. Nicolas Leroux:

Thank you for submitting your manuscript to Microbiology Spectrum. It has been reviewed by two experts, and both viewed the manuscript positively and had some suggestions for revision. I have therefore decided to request 'Modifications', which means that if you choose to, you can submit a revised version of the text following the details outlined below, and that It is likely that I will send the revision out for a second round of review.

You will see that each reviewer has suggested several places where you can clarify the methods or, if necessary, add additional analyses. I generally agree with their concerns. As you revise, I confirm to you that Spectrum's scope is that "rather than making subjective evaluations of potential impact, Microbiology Spectrum publishes research studies that are of high technical quality and are useful to the community." I will be supportive of revisions that meet this expectation, and encourage you to consider these criteria and explain them in your reviewer response during the revision.

<https://spectrum.msubmit.net/cgi-bin/main.plex?el=A7QF1CBMh5A5FRxj3F2A9fdjg991zN7RbO4ZYBFfBH0AZ>

ASM policy requires that data be available to the public upon online posting of the article, so please verify all links to sequence records, if present, and make sure that each number retrieves the full record of the data. If a new accession number is not linked or a link is broken, provide production staff with the correct URL for the record. If the accession numbers for new data are

not publicly accessible before the expected online posting of the article, publication of your article may be delayed; please contact the ASM production staff immediately with the expected release date.

Sincerely,

John Chaston
Editor, Microbiology Spectrum

Journals Department
Reviewer comments:

Reviewer #1 (Comments for the Author):

The paper entitled "Gut microbiota of an Amazonian fish in a heterogeneous riverscape: Integrating Genotype, Environment and Parasitic infections" focused on estimation of different factors that potentially affect on fish gut bacterial community structure. I very like that more and more studies started to include parasites as a potential factor that may modulate gut bacterial community. The present study is very well written, logically presented and statistical analyses were professionally conducted. In the same time, there are several main queries (from my point of view) that have to be addressed. Please see them below.

Lines 72-73. From these, four usually stand out as the most important: environmental conditions, host diet, genotype, and physiological condition.

Comment 1: In the introduction part the authors described environmental conditions, genotype, and physiological condition but nothing was mentioned about the effect of host diet on the bacterial community structure. I guess that such important factor as diet has to be mentioned as well.

Response: Diet was explicitly mentioned in the two previous sentences, lines 69-71: "The taxonomic composition and functional repertoire of the gut microbiota are determined by a wide range of factors. From these, four usually stand out as the most important: environmental conditions, host diet, genotype, and physiological condition".

Lines 175-176. Field trips were conducted from September to December 2018-2019 during the dry season.

Comment 2: How did the authors estimate the potential effect of "year"? The microbiota of aquatic ecosystem can be altered from year to year. The mentioned period (dry season) included four months (September-December). Please clarify the potential effect of different months on the gut bacterial structure and support it by statistical analyses. For example, both March and May belong to "spring" season but various water physico-chemical characteristics could be significantly different between them.

Response: The sampling was planned to minimize any bias due to temporal effect on microbial communities. First, fish and water collection were completed in two months (October, November). Second, both white (WW) and black water (BW) samples were collected the same month in both 2018 and 2019: October 2018: Rio Negro – Anavilhanas (BW), Lago Janauari (WW), Lago Catalão (WW), November 2018: Rio Negro - Barcelos (BW), Lago Janauaca (WW), Rio Manacapuru (WW); October 2019: Rio Branco (WW),; Lago do cemeterio (BW), Rio Negro - Santo Alberto (BW); November 2019: Lago des pirates (WW), Lago Téfê-Solimões (WW), and Lago Téfê (BW). Therefore, we are confident that temporal bias was minimal in differentiating black and white water bacterial and eukaryotic communities. Finally, the potential bias due to temporal effect was assessed by incorporating “Sampling time” in the permanova, which did not detect any effect on microbiota composition.

Lines 195-197. A total of 167 midgut samples were collected and processed. DNA was extracted from 20 mg of a midgut segment for each fish. We specifically selected the middle section of the intestine to be constant in the tissues analysed.

Comment 3: Please clarify the potential effect of content in midgut of fish on general bacterial structure of this gut segment. Did the authors estimate the midgut fullness? It is known that the diversity estimates in gut mucosa and gut content could be different. I guess that among 167 specimens there were fish with different levels of gut fullness.

Response: *M. festivus* is a herbivorous/detritivorous fish which mostly feed on periphyton a resource that is very abundant in Amazonian water in both wet and dry season. For this reason, most fish had their gut full. For this reason, we sliced the same midgut section in each fish including gut content. Including the gut content in the analysis was also favorable for detecting parasitic taxa and describing fish' diet. We did not consider gut fullness in the analysis and considered it a random factor affecting every site at the same rate.

Lines 458-462. For instance, we detected the presence of 135 nonhost-related ASVs from 53 different genera across our 167 *Mesonauta festivus* samples. In comparison, Minich et al. (2018) detected less than 50 Eukaryotic operational taxonomic units (OTUs) in their faeces and gut samples of aquaculture fish.

Comment 4: Sorry, I do not understand why the authors try to find relations between the level of parasite infestation in fish populations from nature and fish cultivated under aquaculture conditions? It is expected that these communities will be very different.

Response: Reviewer is right. The comparable studies are scarce and often use different primers or markers, target other organisms and different environments. We decided to remove this sentence.

Lines 487-501. Note. I very like that the authors discuss about the limitations of DNA-barcoding approach trying to explain observed results. For me, the most important limitation of this approach is impossibility to estimate the "size" of the factor "parasite infestation", I mean, such parameter as intensity level. Indeed, the lack of significant effect of parasite invasion in experiment where available information is categorized only as "yes" or "no" parasites in the fish gut is very vulnerable.

Lines 504-507 and 512-513.

Comment 5: I guess that the most important factor is the present or absent the first intermediated host for given cestode species rather than the type of water. Perhaps, in black water there are no specific zooplanktonic crustations that could be used as the first intermediate hosts by cestodes.

Response: This is a very interesting comment. According to the study of Nakajiima et al. (2017), Rio Negro black water mean density and biomass of mesozooplankton including potential intermediate copepod hosts of cestode were higher compared to the Rio Solimoes white water, by 2.8 and 2.0 times, respectively. Therefore, such results would not support the hypothesis according to which the low prevalence of cestode intermediate copepod host would explain the lower cestode prevalence in fish. However, the taxonomic information provided in Nakajiima et al. (2017) is too limited to state whether water color has an impact on intermediated host for any given cestode species.

Nakajima R, Rimachi EV, Santos-Silva EN, Calixto LSF, Leite RG, Khen A, Yamane T, Mazeroll AI, Inuma JC, Utumi EYK, Tanaka A. The density and biomass of mesozooplankton and ichthyoplankton in the Negro and the Amazon Rivers during the rainy season: the ecological importance of the confluence boundary. *PeerJ*. 2017 May 11;5:e3308. doi: 10.7717/peerj.3308. PMID: 28507821; PMCID: PMC5429737.

Lines 556-558. Similarly, we detected a higher abundance of Cyanobacteria in fish from white water, furthering the potential horizontal transfer of bacteria associated with the higher chlorophyll A concentration in white water sites.

Comment 6: It is good example of horizontal transfer of bacterial taxa from aquatic compartments to fish gut. Similar observation was found by Parshukov et al. 2019. Probably this ms (Supporting Information File S4) will be useful: Variations of the intestinal gut microbiota of farmed rainbow trout, *Oncorhynchus mykiss* (Walbaum), depending on the infection status of the fish. 2019. *Journal of Applied Microbiology* 127(2) DOI: 10.1111/jam.14302.

Response: We thank the reviewer for putting this publication to our attention. We added this sentence (Lines 564-566): “*Dispersal of bacterioplankton taxa from water to fish gut were documented for discus fish (Sylvain et al. 2017), rainbow trout (Parshukov et al. 2019) and Atlantic Salmon (Lavoie et al. 2021).*”.

Lines 412 - 414. However, as these taxa were not detected in the water microbiome sampled in each water type, it was not possible to link these discriminant features to a putative horizontal transfer from the water microbiome. Lines 544-546. Since the gut microbiota of fish is relatively isolated from the water, changes in physicochemical characteristics could have a limited influence on its communities. Lines 572-573. However, we cannot directly link these bacterial strains to the water bacterioplankton as we did not observe these in our water microbiome assessment. Lines 596-598. Overall, our results support a low but present effect of the environmental dissimilarity between black and white water environments on the taxonomic structure of the mid gut microbiota in *M. festivus*.

Comment 7: Due to the author have not data associated with the bacterial community structure of water and fish gut estimated during several time points, low level of relationships between bacterial communities of water and fish gut is expected. Both type of bacterial communities (from water and fish gut) could be characterized by different time that is needed to change them. It could be expected that bacterial community in water is more sensitive and changed faster than the bacterial community in fish gut even if, as the authors pointed out, the gut bacterial community originated from the surrounding water. I understand that sample collection in nature is not the same and easy as in laboratory but the limited data from nature may led to erroneous conclusions.

Response: The reviewer raises another interesting point here. We modified the sentence accordingly (Lines 567-574): “However, it may be hazardous to directly link these bacterial strains to the water bacterioplankton in this study because we did not conduct a time series of sampling to account for a possible decoupling between their respective taxonomic dynamics in water and fish gut.”.

Unfortunately, I did not find any speculations regarding to feeding features of *M. festivus* from black and white waters. The authors mentioned that this fish is detritivorous (line 157) and that host diet is very important factor (line 73) but there are no any analysis or speculations associated with potentially different microbiota of sediments itself in the bottom of white and black waters or differences in community of invertebrates inhabiting the bottom sediments of those places (both sediments and invertebrates may possess by specific microbiomes). That fact that the structure of community of invertebrates are different between black and white waters is partially supported by differences in the structure of fish parasites (as it was shown in the present ms) required invertebrates as the first intermediate hosts (for example, cestodes). Moreover, the authors may have such data because 18S rRNA analysis was conducted.

Comment 8: It could be interesting to compare the gut content of fish from black and white water using 18 S data and, consequently, speculate about similarity of fish diets from studied different sites.

Response: This is a very interesting comment. Here is the reason why we did not add this analysis in the previous version of the manuscript. There are only 18S ASVs potentially representing prey (rotifera, annelidae, arthropoda): 212, 225, 287, 420, 439, 496, 501, 523, 545, 559, 577, 585, 598, 611, 613, 618, 641, 652, 655, 656, 668, 673, 674, 717, 741, 764. However, only 13 gut samples contain at least one read from one of these 26 ASVs. Because these 26 ASVs are detected in a very small fraction of the dataset, we do not think that it is a suitable dataset for testing watercolor effect on diet.

Furthermore, when considering these 13 samples to perform a PERMANOVAs analysis, there were no significant differences between both water types.

```

Permutation test for adonis under reduced model
Terms added sequentially (first to last)
Permutation: free
Number of permutations: 10000

adonis2(formula = (tax_glom(Pruned_18S, taxrank = "genus", NArm = T)@otu_table) ~ Water_type * Genotype, data = data.frame(Pruned_18S@sam_data),
permutations = 10000, method = "bray")

```

	Df	SumOfSqs	R2	F	Pr(>F)
Water_type	1	0.4390	0.07458	0.9051	0.61474
Genotype	3	1.6580	0.28166	1.1395	0.08709
Water_type:Genotype	1	0.3945	0.06701	0.8133	0.93571
Residual	7	3.3951	0.57675		
Total	12	5.8865	1.00000		

```

---
Signif. codes:  0 '****' 0.001 '**' 0.01 '*' 0.05 '.' 0.1 ' ' 1

```

Probably these two papers about microbiota of fish parasites will also be useful.

The effect of diet on the structure of gut bacterial community of sympatric pair of whitefishes (*Coregonus lavaretus*): One story more. 2019PeerJ 7(9) DOI: 10.7717/peerj.8005
 2) Composition of the microbial communities in the gastrointestinal tract of perch (*Perca fluviatilis* L. 1758) and cestodes parasitizing the perch digestive tract. 2019. Journal of Fish Diseases 43(4). DOI:10.1111/jfd.13096

Reviewer #2 (Comments for the Author):

The authors present an interesting study of the gut microbiomes of *M. festivus*, an Amazonian wild fish, and the impact of genotype, water quality, and parasite incidence on the gut microbiome. The manuscript is well-written and then study and analyses are generally sound. That said, I do have several minor comments for the authors to consider:

Comment 1: Line 98: Recommend replacing "horizontal transfer", which is a term that invoked genetic mechanisms, with "dispersal", which invokes the intended ecological mechanism.

Response: The reviewer is right. "horizontal transfer" was replaced with the terms "dispersal" or "recruitment". Lines 95-98: *"Since the surrounding water may be a source of recruitment of environmental microbial strains to the fish gut microbiota (Sylvain et al. 2019, Lavoie et al. 2021), we posed the hypothesis that certain taxa of M. festivus' gut microbiome would be related to the water type (black or white)."*; Lines 543-544: *"could be the result of dispersal from the environment to the microbiota of M. festivus."*

Comment 2: Line 145: I have trouble following the defined scenario here, because it is presented in the context of strong versus weak environmental effects. However, the expectations outlined as a result of these effects includes genetic components. It thus rather seems that the context of the spectrum being defined in this scenario is actually about the strength of the environmental effects relative to the strength of the genetic effects. If that's an accurate interpretation of this scenario, it would help readers to be explicit. If I'm mistaken, clarification in this passage would help.

Response 2: Reviewer is absolutely right. The paragraph was rewritten accordingly. See lines 141-148: *"In any case, the interplay between genetic and environmental effect on gut microbiota taxonomic structure ranges between the following extreme scenario: strong environmental effects relative to weak genetic effects and vice versa. In the first case, fish should display similar microbial community shifts at similar environmental shifts (e.g. independent black water sites versus their respective connected white water sites). On the opposite, in the presence of a weak environmental effect and a strong genetic effect, fish from a same genetic population, but inhabiting contrasting environments, are expected to converge in terms of microbiota composition."*

Comment 3: Line 198: Does the DNeasy kit include mechanical disruption of the samples (e.g., bead beating)? If not, it would be worth articulating whether this could be a caveat impacting the results, as many taxa may be recalcitrant to other forms of cellular lysis.

Response: We did not use mechanical disruption of the samples by bead beating. Instead, samples were cut into small pieces and digested with lysis buffer. We agree that this may have limited the completeness of detection of microbial and parasitic taxonomic diversity. However, this bias remains the same for each sample, which should not affect comparisons between groups. We have included a sentence in the methods to emphasize this point. See lines 192 " *The middle section of the intestine was specifically selected and samples were cut into small pieces using sterile blades and digested with lysis buffer overnight. Although an additional step of*

mechanical disruption would have improved the detection of both microbial and parasitic taxonomic diversity, this potential bias should not affect comparisons between experimental groups. "

Comment 3: Line 222: Please specify the read length and whether the data produced by the MiSeq run was paired end.

Response: This information was provided. See lines 231-236: *"The demultiplexed fastq sequence files were processed through DADA2 (Callahan et al. 2016) using the function "ytfilterAndTrim" with the following parameters: two as the phred score threshold for total read removal, a maximum expected error of two for forward reads and four for reverse reads, a truncation length of 280 base pairs for forward reads and 200 base pairs for reverse reads for 16S rRNA library, and a length of 275 base pairs for forward reads and 250 base pairs for reverse reads for 18S rRNA library."*

This is a pair end sequencing for both 16S and 18S. This is now clarified line 221: *"Multiplex paired-end sequencing was performed using the MiSeq platform"*. We omitted the function *"mergePairs"* for read merging. This is now corrected lines 236-240: *"The filtered reads were then fed to the error rate learning, dereplication, merging and Amplicon Sequence Variant (ASV) inference steps using the functions "learnErrors", "derepFastq" mergePairs and "dada". Chimeric sequences were removed using the "removeBimeraDenovo" function with the "pseudo" method parameter."*

Comment 4: Line 281: Phylogenetic diversity metrics are used, but it is not clear how the phylogeny was assembled. Please provide these details in the Methods.

Response: Phylogenetic diversity was computed with Phyloseq using `estimate_pd(phylo)` function. The tree was computed with the simple agglomerative (bottom-up) hierarchical clustering method (UPGMA). Chao1 and Shannon were computed using `estimate_richness` function. Lines 282-286: *"Using the 16S rRNA Phyloseq object previously produced, we estimated alpha diversity indexes such as the observed number of ASVs (Chao1), and Shannon entropy (Shannon 1949) using estimate_richness function. Then, phylogenetic diversity (Faith 1992) was computed with using estimate_pd(phylo) function. The tree was computed with the simple agglomerative (bottom-up) hierarchical clustering method (UPGMA)."*

Comment 5: Line 323: It isn't clear what is meant by "pooling does not influence the analysis of the results in the discussion".

Response: We clarified the sentence line 328-329 as: *"We previously analysed independently the impact of each type of helminth detected and their pooling did not have any impact on the general significance of the results."*

Comment 6: Line 390: The methods seem to indicate that PERMANOVA was used to evaluate the association between sample covariates and beta-diversity, yet here it seems that an ordination based correlation analysis was used. Can the authors clarify why this approach was used to

address this analysis and not PERMANOVA, which tends to be more robust for these types of investigations since it is not dependent upon the ordination?

Response: PERMANOVA was used to evaluate associations between covariate genotype and water types and the gut microbiome. However, we opted for an ordination-based method for the comparison of the 5 environmental parameters. We have now separated the two paragraphs in the methods to better separate both analyses.

We selected an ordination-based method as it allowed us to share the environmental vectors and the data together as a graphical output highlighting the relative influence of each parameter on the gut microbiome structure. Considering your comment, we decided to run the proposed analysis on a PERMANOVA to make sure we did not miss any important results. Here are the results:

Comparison of both results; Envfit : “ chlorophyll a concentration in water: $R^2 = 0.084$, p-value = 0.001; DOC: $R^2 = 0.003$, p-value = 0.76; Silicate: $R^2 = 0.027$, p-value = 0.11; Conductivity: $R^2 = 0.027$, p-value = 0.11; pH: $R^2 = 0.024$, p-value = 0.14).

PERMANOVA :

```
Permutation test for adonis under reduced model
Terms added sequentially (first to last)
Permutation: free
Number of permutations: 50000

adonis2(formula = (tax_glom(A_16S_Phyloseq, taxrank = "Genus", NArm = T)@otu_table) ~ DOC + Silicate + Chl_a + Conductivity + pH, data =
data.frame(A_16S_Phyloseq@sam_data), permutations = 50000, method = "bray")

```

	Df	SumOfSqs	R2	F	Pr(>F)
DOC	1	0.936	0.01750	3.1197	0.00194 **
Silicate	1	0.701	0.01312	2.3381	0.01574 *
Chl_a	1	0.919	0.01719	3.0636	0.00252 **
Conductivity	1	1.238	0.02315	4.1272	0.00024 ***
pH	1	1.384	0.02589	4.6152	8e-05 ***
Residual	161	48.282	0.90316		
Total	166	53.459	1.00000		

```
---
Signif. codes:  0 '***' 0.001 '**' 0.01 '*' 0.05 '.' 0.1 ' ' 1
```

Results are very different in the PERMANOVA as they show a significant association between each environmental parameter and microbiome beta diversity. However, this is probably due to the very high number of comparisons performed by the PERMANOVA compared to the lower number of comparisons used in envfit which only considers environmental parameters and bray Curtis distances given to the function. Indeed, since this type of analysis cannot be easily plotted and shows all significant results, it is very difficult to highlight a pattern in the data. For these reasons, we think that the envfit analysis, while potentially less powerful in detecting significant interactions, leads to more interesting and interpretable results.

Comment 7: Line 399: Adonis is often found to be sensitive to the order in which the regression covariates are listed. Have the authors validated that reordering genotype and water type in their PERMANOVA regression model produces consistent results to those reported here?

Response: Yes, we previously did the PERMANOVA in both orders to consider this potential bias and, while results are not perfectly identical, they show the same patterns, as exemplified below with one of our main PERMANOVA ran at a low depth.

```

Permutation test for adonis under reduced model
Terms added sequentially (first to last)
Permutation: free
Number of permutations: 20000

adonis2(formula = A_16S_Phyloseq@otu_table ~ Water_type * Genotype, data = data.frame(A_16S_Phyloseq@sam_data), permutations = 20000, method = "bray")
      Df SumOfSqs      R2      F Pr(>F)
Water_type      1   1.376 0.02031 3.5052 5e-05 ***
Genotype         3   2.601 0.03839 2.2088 5e-05 ***
Water_type:Genotype 1   0.571 0.00843 1.4555 0.0575 .
Residual        161  63.205 0.93286
Total           166  67.754 1.00000
---
Permutation test for adonis under reduced model
Terms added sequentially (first to last)
Permutation: free
Number of permutations: 20000

adonis2(formula = A_16S_Phyloseq@otu_table ~ Genotype * Water_type, data = data.frame(A_16S_Phyloseq@sam_data), permutations = 20000, method = "bray")
      Df SumOfSqs      R2      F Pr(>F)
Genotype      3   2.963 0.04374 2.5161 5e-05 ***
Water_type     1   1.014 0.01497 2.5832 0.00050 ***
Genotype:Water_type 1   0.571 0.00843 1.4555 0.06025 .
Residual      161  63.205 0.93286
Total         166  67.754 1.00000
---
Signif. codes:  0 '****' 0.001 '***' 0.01 '**' 0.05 '.' 0.1 ' ' 1

```

Comment 8: Line 567: Are the authors confident in their Cyanobacteria annotations? Could these annotations alternatively result from the presence of chloroplast DNA in the fish diet?

Response: Numerous plants were detected in the gut using 18S marker. However, given that divergence time between Cyanobacteria and the oldest vascular plant fossil (tracheophytes) is about 460 MYA (Kenrick and Crane, 1997), it is very unlikely that Cyanobacteria annotation was corrupted with chloroplast sequences, using E-values threshold as low as 10×10^{-30} .

July 14, 2023

Mx. Nicolas Leroux
Universite Laval
Biology
1030 avenue de la Médecine
Quebec, QC
Canada

Re: Spectrum02755-22R2 (Gut microbiota of an Amazonian fish in a heterogeneous riverscape: Integrating Genotype, Environment and Parasitic infections)

Dear Mx. Nicolas Leroux:

Your manuscript has been accepted, and I am forwarding it to the ASM Journals Department for publication. You will be notified when your proofs are ready to be viewed.

Please be prepared to update the BioProject and SRA #s at that time.

Sincerely,

John Chaston
Editor, Microbiology Spectrum
